# Risk of nontyphoidal *Salmonella* bacteraemia in African children is modified by *STAT4*

James J. Gilchrist [1,2], Anna Rautanen[1], Benjamin P. Fairfax[1], Tara C. Mills[1], Vivek Naranbhai [1], Holly Trochet[1], Matti Pirinen [1,3], Esther Muthumbi[4], Salim Mwarumba[4], Patricia Njuguna[4], Neema Mturi[4], Chisomo L. Msefula[5,6], Esther N. Gondwe[5], Jenny M. MacLennan[5,7], Stephen J. Chapman[1,8], Malcolm E. Molyneux[5], Julian C. Knight[1], Chris C.A. Spencer[1], Thomas N. Williams[4,9], Calman A. MacLennan[5,10], J. Anthony G. Scott [4,11] & Adrian V.S. Hill [1,10]

Nontyphoidal *Salmonella* (NTS) is a major cause of bacteraemia in Africa. The disease typically affects HIV-infected individuals and young children, causing substantial morbidity and mortality. Here we present a genome-wide association study (180 cases, 2677 controls) and replication analysis of NTS bacteraemia in Kenyan and Malawian children. We identify a locus in *STAT4*, rs13390936, associated with NTS bacteraemia. rs13390936 is a context-specific expression quantitative trait locus for *STAT4* RNA expression, and individuals carrying the NTS-risk genotype demonstrate decreased interferon-γ (IFNγ) production in stimulated natural killer cells, and decreased circulating IFNγ concentrations during acute NTS bacteraemia. The NTS-risk allele at rs13390936 is associated with protection against a range of autoimmune diseases. These data implicate interleukin-12-dependent IFNγ-mediated immunity as a determinant of invasive NTS disease in African children, and highlight the shared genetic architecture of infectious and autoimmune disease.

[1] Wellcome Trust Centre for Human Genetics, University of Oxford, Oxford OX3 7BN, UK. [2] Department of Paediatrics, University of Oxford, Oxford OX3 9DU, UK. [3] Institute for Molecular Medicine Finland (FIMM) University of Helsinki, FI-00014 Helsinki, Finland. [4] KEMRI-Wellcome Trust Research Programme, Kilifi 80108, Kenya. [5] Malawi-Liverpool-Wellcome Trust Clinical Research Programme, College of Medicine, P.O. Box 30096, Chichiri, Blantyre, Malawi. [6] Pathology Department, College of Medicine, P.O. Box 360, Chichiri, Blantyre, Malawi. [7] Department of Zoology, University of Oxford, South Parks Road, Oxford OX1 3PS, UK. [8] Oxford Centre for Respiratory Medicine, Churchill Hospital Site, Oxford University Hospitals, Oxford OX3 7LE, UK. [9] Department of Medicine, Imperial College, Norfolk Place, London W2 1PG, UK. [10] The Jenner Institute, University of Oxford, Old Road Campus Research Building, Oxford OX3 7DQ, UK. [11] Department of Infectious Disease Epidemiology, London School of Hygiene and Tropical Medicine, Keppel Street, London WC1E 7HT, UK. Correspondence and requests for materials should be addressed to J.J.G. (email: james.gilchrist@well.ox.ac.uk) or to A.V.S.H. (email: adrian.hill@ndm.ox.ac.uk)

Nontyphoidal *Salmonella* (NTS) is a common, and frequently fatal, cause of bacteraemia in children and HIV-infected adults in sub-Saharan Africa[1]. NTS is estimated to cause 1.9 million episodes of invasive disease resulting in 390,000 deaths annually in Africa[2]. This burden of disease reflects inadequate control strategies for NTS in sub-Saharan Africa, with expanding antibiotic resistance[3], and no anti-NTS vaccine available for use in humans[4]. There is marked inter-individual variation in susceptibility to invasive NTS (iNTS) disease. Both heritable and acquired risk factors for disease are well documented, with sickle cell disease[5], HIV infection, malnutrition[6] and malaria[7] all contributing to susceptibility to iNTS disease in African children.

Genome-wide association studies (GWAS) have been highly successful in identifying common genetic variants associated with hundreds of disease traits in diverse populations[8], including infectious diseases in African populations[9]. We have previously conducted a GWAS of all-cause bacteraemia in Kenyan children, in which we were unable to identify genetic determinants of invasive bacterial disease beyond sickle cell disease, an association we observed across a broad range of bacterial pathogens, including NTS[10]. Despite a smaller sample size, an analysis of pneumococcal bacteraemia within the same collection, however, did demonstrate evidence of novel genetic associations at genome-wide significance[10], suggesting that in this context the loss of study power resulting from a substantial reduction in case numbers is more than offset by the provision of a more precisely defined phenotype. This observation and the lack of unbiased, population-based data assessing host genetic susceptibility to iNTS disease in African populations led us to conduct a GWAS of NTS bacteraemia in the Kenyan collection[10] with additional replication in a Malawian case–control sample collection.

In this study we report a GWAS of susceptibility to NTS bacteraemia in African children. We identify a susceptibility locus for NTS bacteraemia at signal transducer and activator of transcription 4 (*STAT4*), rs13390936. This NTS-associated genetic variation is a context-specific, expression quantitative trait locus (eQTL) for *STAT4* RNA expression in stimulated leucocytes, where the NTS-associated genotype results in reduced *STAT4* RNA expression. We demonstrate that carriage of the NTS-associated genotype at *STAT4* is associated with reduced IFNγ protein production in stimulated natural killer (NK) cells, and with reduced serum IFNγ levels in African children with iNTS infection. Delivery of an effective anti-NTS vaccine is urgently required. By establishing the importance of IFNγ-mediated immunity in the control of NTS infection in African children, our data highlight the need to consider the induction of cell-mediated immunity in the design and evaluation of such a vaccine.

## Results

**NTS GWAS and replication analysis.** Following quality control (QC; Supplementary Tables 1 & 2) and genome-wide imputation, 2847 Kenyan samples (180 cases, 2677 controls) were included in the GWAS discovery analysis of 5,585,198 (additive model) and 4,669,480 (genotypic model) autosomal single-nucleotide polymorphisms (SNPs). Inspection of quantile–quantile plots (Supplementary Fig. 1) and the genomic control inflation factors ($\lambda_{additive} = 1.01$, $\lambda_{genotypic} = 1.01$) indicate that inclusion of the four major principal components (Supplementary Fig. 2) as covariates in the analysis adequately controls for population substructure. In the discovery analysis, 67 SNPs at 16 loci were suggestively associated ($P < 1 \times 10^{-6}$) with NTS bacteraemia (Supplementary Table 3 and Supplementary Fig. 3). SNPs in each of these associated loci were selected for genotyping in Kenyan and Malawian

replication samples. Imputation accuracy of SNPs taken forward for replication genotyping was confirmed in 930 of the Kenyan discovery samples (33% of the discovery samples; 180 cases, 750 controls) by Sequenom MASSArray (mean $r^2 = 0.97$, Supplementary Table 4).

Following the discovery analysis, an additional 1374 Kenyan samples (38 cases, 1336 controls) and 489 Malawian samples (150 cases, 339 controls) were included in the replication genotyping (Supplementary Table 4). In the replication analysis, an intronic SNP (rs13390936, chr2:191954816) in the *STAT4* gene demonstrated evidence of association with NTS bacteraemia in both the Kenyan and Malawian replication samples (Fig. 1 and Table 1). Comparison of the NTS association at rs13390936 under alternative genetic models demonstrated that the observed association is best explained by a recessive model (Supplementary Table 5), with carriage of the minor, TT genotype being associated with increased risk of NTS bacteraemia. Re-analysis of the *STAT4* locus in the Kenyan discovery samples (181 cases, 2759 controls; this includes samples excluded from the main analysis for relatedness) using a linear mixed model, to account for population structure and relatedness, confirms the association of rs13390936 genotype with NTS bacteraemia (recessive model; $P = 7.14 \times 10^{-7}$).

Re-imputation of the *STAT4* region, with 1000G Phase 3 as a reference panel, confirmed rs13390936 as the variant most strongly associated with NTS in both the Kenyan discovery and replication samples (Fig. 1). A Bayesian analysis of the re-imputed data demonstrated that a set of seven SNPs, including rs13390936, has a >95% probability of containing the causal variant, under the assumption that we have accurately imputed the single causal SNP in our data (Fig. 1 and Supplementary Table 6). To more fully confirm imputation accuracy at rs13390936 in the Kenyan discovery samples, we re-genotyped rs13390936 in the complete sample collection with high-resolution melting curve analysis (HRMA). HRMA and imputation concordance at rs13390396 confirmed imputation accuracy ($r^2 = 0.97$), and confirmed the association with NTS bacteraemia (Table 1). In the combined analysis of all samples, under a recessive model, the association of rs13390936 with NTS bacteraemia exceeds genome-wide significance (fixed-effects meta-analysis, $P = 8.62 \times 10^{-10}$, odds ratio (OR) 7.61, 95% confidence interval (CI), 3.98–14.55).

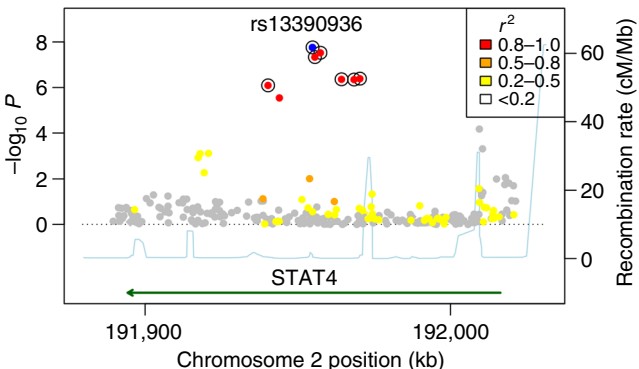

**Fig. 1** Association plot of NTS bacteraemia susceptibility at the *STAT4* region in Kenyan children. NTS association at the *STAT4* region under the recessive model in Kenyan discovery samples ($n = 180$ cases, 2677 controls). rs13390936 is highlighted in blue. SNPs are coloured according to the strength of linkage disequilibrium ($r^2$) to rs13390936. SNPs in a credible SNP set, which includes the causal variant with >95% probability, are ringed with black circles

**Table 1 Association of rs13390936 with NTS bacteraemia in African children**

| Population | Case | | Control | | Genotypic model | Recessive model | |
|---|---|---|---|---|---|---|---|
| | Freq. | TT/AT/AA | Freq. | TT/AT/AA | P value | P value | OR (95% CI) |
| Kenya (discovery) | 0.151 | 8/40/138 | 0.107 | 15/529/2068 | $1.76 \times 10^{-5}$ | $3.33 \times 10^{-6}$ | 8.03 (3.34–19.31) |
| Kenya (replication) | 0.146 | 3/4/30 | 0.106 | 12/254/1053 | $1.83 \times 10^{-3}$ | $6.54 \times 10^{-4}$ | 9.96 (2.66–37.36)* |
| Malawi | 0.15 | 6/31/106 | 0.136 | 3/86/249 | 0.039 | 0.026 | 4.89 (1.21–19.83)* |
| Combined | | | | | | $8.62 \times 10^{-10}$ | 7.61 (3.98–14.55) |

rs13390936 is directly genotyped in all samples by HRMA. Freq. minor allele frequency, OR odds ratio, CI confidence interval, HRMA high-resolution melting curve analysis
*Fisher's exact estimates of effect sizes in the Kenyan (OR = 9.56, $P = 6.79 \times 10^{-3}$) and Malawian (OR = 4.87, $P = 0.023$) replication samples (without covariates) are comparable to those derived by logistic regression

**Established NTS-risk factors and rs13390936**. Malaria, HIV infection and malnutrition are important, acquired risk factors for NTS bacteraemia in African children[6,7]. To address whether the observed association of rs13390936 with NTS bacteraemia is independent of these acquired risk factors, we conducted a Bayesian analysis comparing models of association at rs13390936 with NTS bacteraemia with and without acquired risk factors for NTS in the Kenyan discovery samples (Supplementary Fig. 4). In that analysis, the most probable models are those in which rs13390936 is associated with susceptibility to NTS bacteraemia with the same effect size in children with and without each of malaria and malnutrition. The numbers of HIV-infected children in the Kenyan discovery samples ($n = 24$, none of which carry the rs13390936:TT genotype) are too small to permit a stratified analysis. To ensure HIV infection did not confound the observed association between rs13390936 genotype and NTS bacteraemia, we repeated the association analysis at rs13390936 including only HIV-uninfected NTS cases ($n = 97$). In that analysis, the observed effect size (recessive model; OR 8.49, 95% CI, 2.37–24.28) is in keeping with that observed in the association analysis including all samples (Table 1), further supporting the association between rs13390936 genotype and NTS bacteraemia.

Control samples used in the Kenyan discovery analysis are taken from a birth cohort, and acquired risk factor data are not available to perform a regression analysis of NTS bacteraemia risk adjusted for malaria, malnutrition and HIV. To address this limitation, we fitted a regression model of rs13390936 association with NTS bacteraemia in the Malawian replication samples, including HIV infection, malaria parasitaemia and severe malnutrition as covariates. In that model, including 396 Malawian children (109 cases, 287 controls) for which complete covariate data are available, rs13390936 is associated with NTS bacteraemia under a recessive model (logistic regression, $P = 9.96 \times 10^{-3}$; OR 7.74, 95% CI, 6.18–9.30) independent of HIV, malaria and malnutrition. This lack of confounding by acquired risk factors for NTS is in keeping with the absence of observed association between genetic variation at the *STAT4* locus or HIV and malaria-related phenotypes in GWAS published to date[11,12]. To further address the possibility of confounding secondary to malaria in Kenyan and Malawian children, we used publically available GWAS summary statistics from the MalariaGEN consortium GWAS of severe malaria in Gambian, Kenyan and Malawian children[12] (total 5126 cases, 5287 controls; Gambia 2428 cases, 2489 controls; Malawi 1193 cases, 1321 controls; Kenya 1505 cases, 1477 controls) to assess for any evidence of association between rs13390936 and severe malaria. In that dataset, rs13390936 is not associated with severe malaria in African children (logistic regression; additive model, $P = 0.31$, OR 0.96, 95% CI, 0.88–1.04; dominant model $P = 0.47$, OR 0.97, 95% CI, 0.88–1.06; recessive model $P = 0.15$, OR 0.76, 95% CI, 0.52–1.11; heterozygous advantage model $P = 0.98$, OR 0.98, 95% CI, 0.89–1.08).

Rates of HIV and malaria co-infection, but not malnutrition, among Kenyan and Malawian children with NTS included in the study are significantly different (Supplementary Table 7). The observed effect size of the rs13390936 association with NTS bacteraemia is lower in Malawian children than that observed in Kenyan children, albeit with no evidence of inter-study heterogeneity of effect (Table 1). While there is no evidence that the observed association between rs13390936 genotype and NTS bacteraemia is the result of confounding secondary to acquired risk factors for NTS, we further investigated whether there is evidence that the effect of rs13390936 genotype on NTS disease risk is modified by HIV, malaria or malnutrition. We fitted regression models of NTS bacteraemia and rs13390936 association, with and without interaction terms between rs13390936 and each acquired risk factor, in the same set of Malawian children. In that analysis, there is no evidence for interaction between carriage of the rs13390936:TT genotype and HIV status (likelihood ratio test, $P = 0.81$), malaria co-infection (likelihood ratio test, $P = 0.07$) or malnutrition (likelihood ratio test, $P = 0.82$) in NTS bacteraemia risk.

The sickle cell locus (rs334) has been previously demonstrated to be associated with increased risk of NTS bacteraemia in Kenyan children[5] (sickle cell disease—HbSS), and with protection against malaria, and as a consequence, bacteraemia[7] (sickle cell trait—HbAS). In the Kenyan discovery samples (164 cases, 2342 controls), sickle cell disease is associated with increased risk of NTS bacteraemia (logistic regression, $P = 8.30 \times 10^{-5}$; OR 4.89, 95% CI, 2.10–10.40), and there is no statistically significant evidence (logistic regression, $P = 0.65$) for an effect of sickle cell trait (HbAS) on risk of NTS bacteraemia (Supplementary Fig. 5). As carriage of HbSS is associated with risk of NTS bacteraemia, and carriage of HbAS is associated with a key risk factor for NTS bacteraemia (malaria), we sought to assess whether the observed association at the *STAT4* locus is independent of genotype at rs334. In regression models conditioned on genotype at rs334 (Supplementary Fig. 5), the observed association at rs13390936 with NTS bacteraemia persists when conditioned on HbAS (OR 7.16, 95% CI, 2.68–17.41) and HbSS (OR 8.08, 95% CI, 2.71–21.89).

**rs13390936 is associated with *STAT4* RNA expression**. GWAS-identified trait-associated loci are enriched for regulatory DNA elements[13]. Given rs13390936 is non-coding, we explored whether it is associated with gene expression. We analysed the effects of rs13390936 genotype on *STAT4* RNA expression, using eQTL datasets from previously published and unpublished datasets[14–16] of naïve and stimulated primary immune cells from healthy European adults. To investigate whether this analysis could be confounded by differential linkage disequilibrium (LD) between European and African populations, we examined LD patterns at the NTS-associated locus in individuals of African and European ancestry included in the study. In that analysis, there is no

evidence of differential LD at the NTS-associated locus between the Kenyan discovery GWAS samples and the European samples used in the eQTL analyses (Supplementary Fig. 6). In naïve immune cells, rs13390936 genotype was not significantly associated with *STAT4* RNA expression, which was most abundant in NK cells (Fig. 2a). By contrast, rs13390936 was a significant expression quantitative trait locus (eQTL) for *STAT4* RNA expression in monocytes following stimulation with

lipopolysaccharide for 2 (linear regression, $P = 4.52 \times 10^{-5}$) or 24 h (linear regression, $P = 2.65 \times 10^{-5}$), or IFNγ for 24 h (linear regression, $P = 2.63 \times 10^{-5}$), with carriers of the NTS-risk genotype expressing the least *STAT4* in each stimulation condition (Fig. 2b).

**rs13390936 is associated with IFNγ production in NK cells.** There is no established role for STAT4 in monocytes during

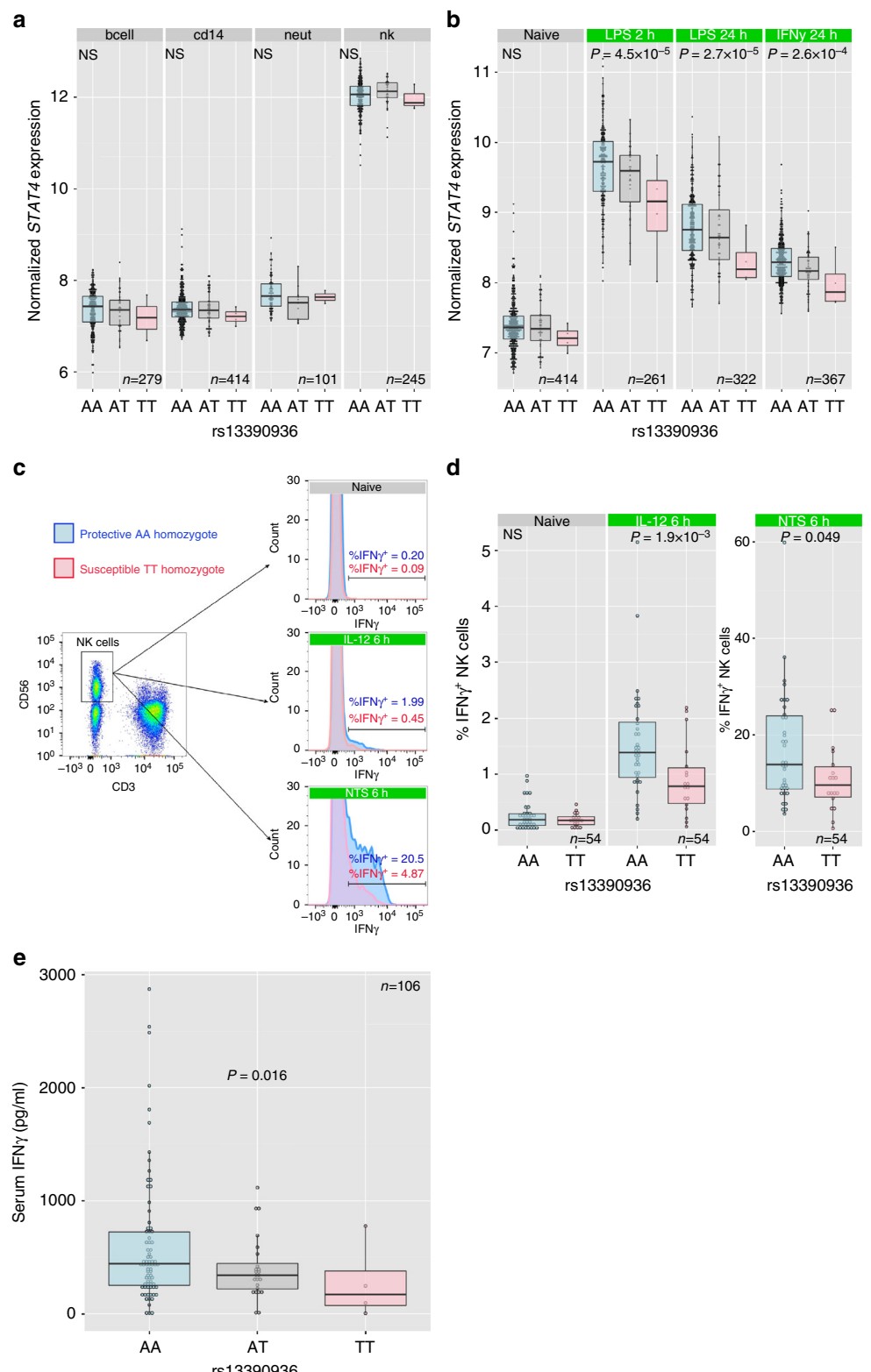

*Salmonella* infection. By contrast, a major consequence of STAT4 activity is IFNγ production in NK cells and CD4[+] T cells. To better understand the NTS-associated phenotype of rs13390936 at the protein level in these cells, we recruited 54 healthy European adults of known genotype at rs13390936 (36 donors carrying the NTS-protective AA genotype; 18 carrying the NTS-susceptible TT genotype) via a genotype-selectable bioresource (Oxford Biobank). In these individuals, we investigated genotype-dependent IFNγ production in naïve and stimulated NK cells and CD4[+] T cells by intracellular cytokine staining and flow cytometry. IFNγ production was quantified at 6 h in naïve cells and in cells following stimulation with IL-12 and NTS (Fig. 2c). Donors with the NTS-risk rs13390936 genotype produced a reduced fraction of IFNγ[+] NK cells following stimulation with IL-12 (linear regression, $P = 1.91 \times 10^{-3}$, mean$_{AA}$ = 1.54%, mean$_{TT}$ = 0.91%) or NTS (linear regression, $P = 0.049$, mean$_{AA}$ = 17.09%, mean$_{TT}$ = 10.76%) (Fig. 2d). We observed this genotype-dependent effect in NK cells but not in CD4[+] T cells (Supplementary Fig. 7).

**rs13390936 is associated with IFNγ in acute NTS bacteraemia**. To extend our understanding of the IFNγ protein phenotype of rs13390936 to African children and to determine the effect of rs13390936 on total IFNγ production during episodes of NTS bacteraemia, we measured serum IFNγ levels in Malawian children admitted to hospital with NTS bacteraemia ($n = 106$) during the acute phase of disease and correlated this with genotype at rs13390936 (Fig. 2e). There is no evidence for association between genotype at rs13390936 and co-morbidity, clinical features or outcome in these children (Supplementary Table 8). In keeping with our results in stimulated NK cells in European adults, serum IFNγ levels in Malawian children with acute NTS bacteraemia are dependent on rs13390936 genotype (linear regression, $P = 0.016$), independent of age, sex, malnutrition, HIV status and malaria co-infection. Malawian children carrying the NTS-risk genotype have reduced circulating IFNγ concentrations (mean$_{AA}$ = 613.9 pg/ml, mean$_{TT}$ = 281.4 pg/ml). Given the importance of CD4[+] T cells as a source of IFNγ, to ensure HIV infection did not confound this association, we also performed this analysis including only HIV-uninfected children ($n = 61$). This confirms the association between serum IFNγ levels during acute NTS bacteraemia and rs13390936 genotype (linear regression, $P = 0.038$, Supplementary Fig. 8). In HIV-infected children ($n = 43$), there is no evidence for association between serum IFNγ levels during acute NTS bacteraemia and rs13390936 genotype (linear regression, $P = 0.23$, Supplementary Fig. 8). The lack of association between rs13390936 genotype and serum IFNγ may simply reflect a lack of power in HIV-infected children, or may reflect other sources of variation in IFNγ response in HIV-infected children (e.g. CD4[+] T cell count).

**Bacteraemia risk conferred by rs13390936 is specific to NTS**. To investigate whether the susceptibility to bacteraemia associated with rs13390936 is specific to NTS or results in susceptibility to other causes of bacteraemia, we conducted a Bayesian analysis comparing models of association at rs13390936 with the major causes of bacteraemia in the Kenyan discovery samples (Fig. 3a, b). In that analysis, the most probable model is one in which rs13390936 is associated with susceptibility to NTS bacteraemia but not bacteraemia caused by other pathogens. This suggests that bacteraemia susceptibility conferred by genetic variation at STAT4 is specific to NTS in this context.

Again using a Bayesian approach to compare models of association at rs13390936, we further investigated whether the increased risk of NTS bacteraemia conferred by rs13390936 genotype is specific to the two major NTS serovars causing disease in African children, *S*. Typhimurium and *S*. Enteritidis. In that analysis, in the Kenyan discovery samples, the most probable model is one in which rs13390936 is associated with susceptibility to bacteraemia secondary to both *S*. Typhimurium and *S*. Enteritidis (Supplementary Fig. 9). This suggests, in keeping with these serovars causing clinically indistinguishable syndromes, and possessing the same acquired risk factors[17], that host genetic risk factors for NTS bacteraemia are shared by both of the major causative serovars.

**rs13390936 is associated with risk of autoimmune disease**. Previously published genetic association studies have implicated variation at other loci within STAT4 in the pathogenesis of several autoimmune diseases[18–20]. To assess the evidence for any shared genetic aetiology between NTS bacteraemia and autoimmune disease at rs13390936, we estimated effect sizes and 95% confidence intervals for association at the NTS-associated locus with eight autoimmune diseases from summary statistics in the ImmunoBase database[21]. We observed a protective association between the NTS-risk allele and a range of autoimmune diseases, with the strongest evidence of association seen with type 1 diabetes, coeliac disease, juvenile idiopathic arthritis, primary biliary cirrhosis and rheumatoid arthritis (Fig. 3c). The direction of this effect suggests that, in contrast to its role in NTS-risk, increased STAT4 expression and IFNγ production enhances the risk of a range of autoimmune diseases.

**Discussion**

In this study, we demonstrate that genetic variation in STAT4 is a determinant of NTS bacteraemia in African children. The NTS-associated SNP in STAT4 (rs13390936) is intronic, and is associated with disease under a recessive model with a large effect size. In common with many examples of trait-associated genetic variation identified by GWAS[13], rs13390936 acts as, or is in LD with, a regulatory determinant of STAT4 expression. Regulatory genetic variation is commonly specific to cell type and context[14], and this is the case for rs13390936 and STAT4. The role of rs13390936 as an eQTL for STAT4 RNA in immune cells is only evident following innate immune stimulation. Moreover, the downstream

**Fig. 2** RNA and protein phenotypes of rs13390936 genotype. Genotype–phenotype correlations are analysed by regression and analysis of variance. *P* values are calculated by F-tests with one degree of freedom. **a** rs13390936 genotype is not significantly associated with STAT4 RNA expression in unstimulated B cells, monocytes, neutrophils or NK cells. **b** Healthy European adults carrying the NTS-susceptibility (T) allele have reduced STAT4 expression following LPS (2 and 24 h) and IFNγ (24 h) stimulation in monocytes. **c** Gating strategy for IFNγ [+] NK cells. Representative IFNγ expression in naïve, and IL-12-simulated or NTS-simulated whole blood is shown for healthy European adult donors carrying the NTS-protective AA genotype (blue) and the NTS-susceptible TT genotype (red). **d** Healthy European adults with the NTS-susceptible (TT) genotype have significantly fewer IFNγ[+] NK cells at 6 h following IL-12 (mean$_{AA}$ = 1.54%, mean$_{TT}$ = 0.91%) or NTS (mean$_{AA}$ = 17.09%, mean$_{TT}$ = 10.76%) stimulation. rs13390936 genotype does not significantly perturb IFNγ[+] NK cell numbers in unstimulated whole blood (mean$_{AA}$ = 0.25%, mean$_{TT}$ = 0.19%). **e** Malawian children (total $n = 106$, 97 with complete covariate data included in final model; rs13390936: AA, $n = 69$; TA, $n = 24$; TT, $n = 4$) with the NTS-susceptibility (T) allele have reduced circulating levels of IFNγ during acute NTS bacteraemia (mean$_{AA}$ = 613.9 pg/ml, mean$_{TT}$ = 281.4 pg/ml) in a linear model adjusted for age, sex and NTS-associated co-morbidity (HIV infection, malnutrition and malaria). Box and whisker plots; boxes depict the upper and lower quartiles of the data, and whiskers depict the range of the data excluding outliers (outliers are defined as data-points >1.5× the inter-quartile range from the upper or lower quartiles)

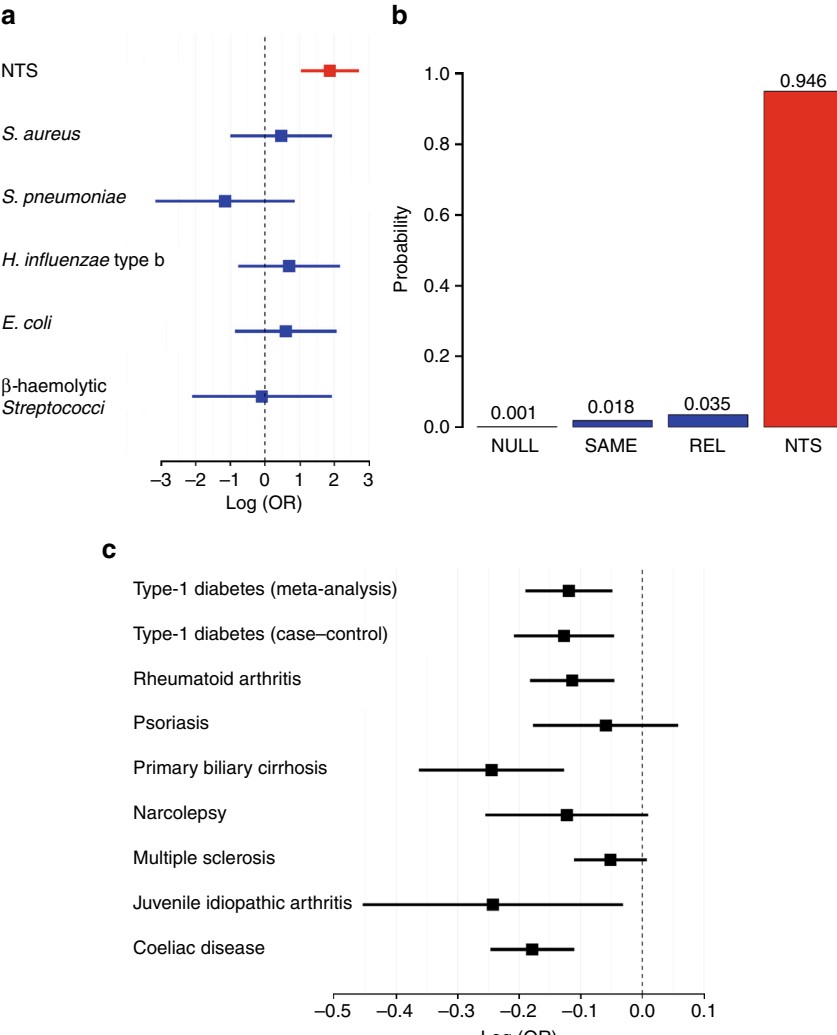

**Fig. 3 a** *STAT4* association with major bacterial pathogens in Kenyan children. Log-transformed odds ratios and 95% confidence intervals of rs13390936 association (recessive model) in Kenyan discovery samples. **b** Posterior probabilities of models of association at rs13390936: NULL, no association with any bacterial pathogen; SAME, the same effect across all bacterial pathogens; REL, related effects across all bacterial pathogens; NTS, a non-zero effect in NTS alone. In a comparison of all possible combinations of rs13390936 association with these pathogens, the model in which NTS alone (highlighted in red) is associated with rs13390936 is the most probable (Bayes factor cf. NULL = 1327). **c** *STAT4* association with autoimmune disease in populations of European ancestry. Log-transformed odds ratios and 95% confidence intervals of rs16833239 (in perfect linkage disequilibrium with rs13390936 in European populations; $r^2 = 1$, $D' = 1$) association (additive model) with autoimmune disease in European populations

effects of that regulatory variation at the protein level exhibit both cell type and context specificity, with rs13390936 genotype modulating IFNγ production capacity in NK cells, but not CD4+ T cells. We further demonstrate that carriage of the NTS-risk allele at rs13390936 associates with reduced IFNγ levels in the serum of African children during acute iNTS disease.

STAT4 is a member of the STAT family of transcription factors[22]. In NK cells and CD4+ T cells, STAT4 phosphorylation in response to IL-12 signalling results in type 1 T helper ($T_H$1) differentiation and IFNγ production (Fig. 4). The importance of this pathway in the control of NTS infection in humans has been established by studies of a group of genetically heterogeneous, rare primary immunodeficiencies, characterized by extreme susceptibility to poorly pathogenic mycobacteria and NTS, collectively designated Mendelian susceptibility to mycobacterial disease (MSMD)[23–29]. While mutations in *STAT4* have not been reported as a cause of MSMD, the described MSMD loci are all in genes with roles in IL-12-dependent IFNγ production and IFNγ-mediated immunity (Fig. 4). The role of STAT4 in host immunity

to NTS infection is further supported by the enhanced susceptibility of Stat4-deficient mice to systemic NTS infection[30]. The identification of regulatory genetic variation affecting IL-12-dependent IFNγ-mediated immunity as a risk factor for NTS bacteraemia highlights the common biological determinants of susceptibility to infectious diseases at the population level and in individuals with rare primary immunodeficiencies.

The identification of genetic variation modifying IFNγ production in NK cells, but not CD4+ T cells, as a risk factor for NTS bacteraemia is noteworthy. These observations are in keeping with findings in the mouse model of NTS infection, in which NK cells have been shown to have a protective role in anti-*Salmonella* immunity, in an IFNγ-dependent manner[31]. The observed association between NTS-associated genetic variation, NK cell IFNγ production and serum IFNγ production in acute NTS bacteraemia suggests that NK cells are an important source of IFNγ in NTS infections in African children. The lack of a similar association between NTS-associated genetic variation and CD4+ T cell IFNγ production does not preclude an important role for CD4+ T

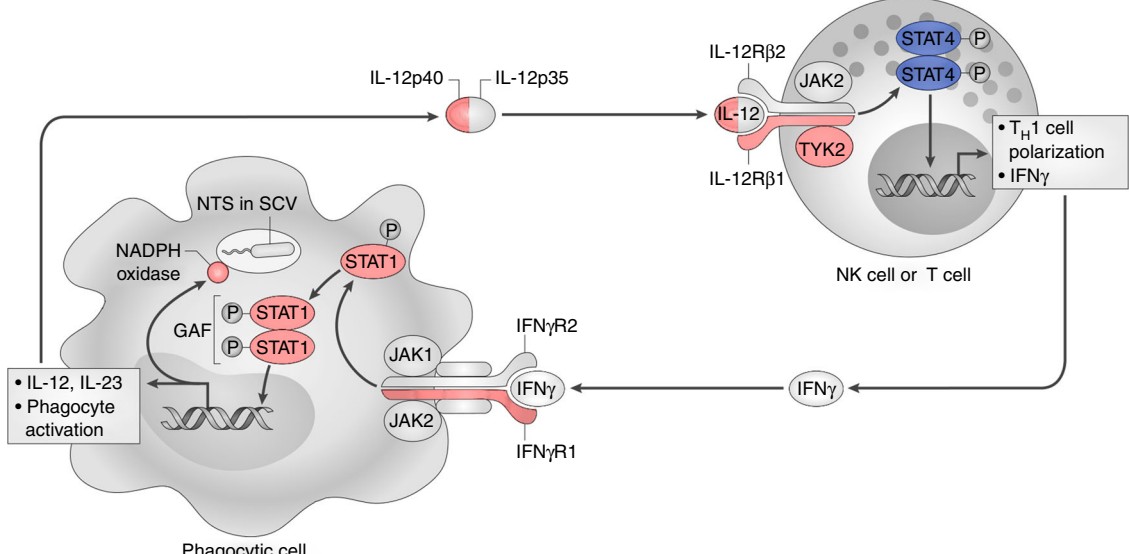

**Fig. 4** The role of STAT4 in the control of intracellular *Salmonella* infection. Internalization of NTS by a phagocyte, in a *Salmonella*-containing vacuole (SCV), results in interleukin-12 (IL-12) release. IL-12 signals via the IL-12 receptor complex on NK cells and T cells leading to the phosphorylation and activation of STAT4 (blue). Activated STAT4 drives IFNγ production from both T cells and NK cells, and $T_H1$ polarization. Released IFNγ activates infected phagocytes via the IFNγ receptor, resulting in the phosphorylation and homodimerization of STAT1 to form γ-activating factor (GAF), which upregulates anti-bacterial effector mechanisms. Genes causing Mendelian susceptibility to mycobacterial disease (MSMD) in which susceptibility to invasive NTS infection has been described are highlighted in pink. (Figure modified from reference [29], first published in *Nature Reviews Immunology*, volume 15, pages 452–463, 2015, by Nature Publishing Group)

cell-derived IFNγ in anti-*Salmonella* immunity in African children. It is, however, consistent with the observation that acquisition of *Salmonella*-specific CD4+ T cells in Malawian children does not result in effective anti-*Salmonella* immunity[32].

A striking feature of MSMD is the narrow and specific range of pathogens to which affected individuals are susceptible. In keeping with this, we demonstrate that bacteraemia susceptibility conferred by genetic variation at *STAT4* is specific to NTS among the most frequent causes of bacteraemia in Kenyan children. By contrast, we observe a protective effect of the NTS-risk allele at rs13390936 across a range of autoimmune diseases. The observation that NTS shares a genetic susceptibility locus with a variety of autoimmune diseases is consistent with previous examples of a shared genetic architecture between autoimmune and infectious diseases[33], and supports the hypothesis that selection pressure imposed by infectious agents has impacted the evolution of autoimmune disease.

A consequence of our use of a birth cohort as the control population for the Kenyan samples in this study is that we were unable to correct for transient risk factors (e.g. malaria and malnutrition) in the discovery analysis. We demonstrate that risk of NTS bacteraemia conferred by rs13390936 genotype is independent of malaria, malnutrition and HIV status in Malawian children. It is important to recognize, that while our analysis provides no evidence for acquired NTS-risk factors modifying the effect of rs13390936 genotype on risk of NTS disease, this study has limited power to detect such an interaction. It will therefore be important to establish in future studies, whether the association between rs13390936 and risk of NTS bacteraemia is observed consistently across populations of African children with diverse rates of HIV, malaria and malnutrition. The limited sample size, in particular with respect to case numbers, restricted this analysis to the contribution of common genetic variation (minor allele frequency >10%) to risk of NTS bacteraemia. Larger sample sizes will be required to assess the contribution of less common genetic

variants, and variants with smaller effect sizes, on the risk of NTS bacteraemia in African children.

Taken together, these results identify regulatory genetic variation in *STAT4* as a key determinant of susceptibility to NTS bacteraemia in African children. We identify IL-12-dependent IFNγ production in NK cells as a mechanism by which genetic variation in *STAT4* modifies risk of NTS disease. In future studies it will be important to establish whether NTS-associated genetic variation also perturbs $T_H17$ responses and IFNγ production in other cell types. Our data have important implications for the design of an anti-NTS vaccine for African children, supporting the development of vaccine candidates that induce robust cell-mediated immunity to NTS.

## Methods

**NTS GWAS study cohorts**. For the Kenyan study cohort, children (under 13 years of age) presenting to Kilifi District Hospital, Kenya between 1 August 1998 and 30 October 2010 with all-cause bacteraemia were recruited[10]. Blood samples for bacterial culture (BACTEC 9050, Becton Dickinson) were taken from every child admitted to the hospital during the study period, with the exception of elective surgical admissions and children admitted following minor accidents. Control samples were collected as part of a birth cohort study from consecutive births between 1 May 2006 and 30th April 2008, among the same population as the case samples in Kilifi district. All control children were recruited under the age of 12 months. Longitudinal follow-up of the control children suggests that the risk of case–control misclassification is negligible (cases of bacteraemia among controls during follow-up = 12; mortality among controls during follow-up = 49). These case and control samples were divided into discovery (cases = 1885, controls = 3000) and replication (cases = 532, controls = 1444) sets for the Wellcome Trust Case Control Consortium 2 all-cause bacteraemia GWAS[10].

From this collection, we extracted children with NTS bloodstream infection (*n* = 267) from the bacteraemia cases, and all of the control children (*n* = 4444) to perform a genome-wide association analysis of NTS bacteraemia in Kenyan children. NTS serotyping was performed according to the Kauffman–White scheme using commercial antisera. As the discovery and replication sample sets had been genotyped using different platforms (see below), we analysed the discovery (cases = 218, controls = 3000) and replication sets separately (*n* = 49, controls = 1444). Demographics of study cases and controls in both the discovery and replication sample sets are presented in Supplementary Table 9. The

frequencies of NTS serotypes and co-morbidities in discovery and replication case samples is presented in Supplementary Table 7. Following explanation of the study, written informed consent was obtained from the parent or guardian of each child included in the study. Ethical approval for study was obtained from the Kenya Medical Research Institute (KEMRI) National Scientific Steering and Research Committees and the Oxford Tropical Research Ethics Committee.

For the Malawian replication study cohort, case samples ($n = 150$) were recruited from children following admission to Queen Elizabeth Central Hospital, Blantyre, Malawi in 2006. Bacterial culture of blood samples was performed using a BacT/Alert 3D system (bioMérieux) in all children admitted with suspected sepsis. Identification of NTS was performed using API 20E kits (bioMérieux) and NTS serotyping performed as for the Kenyan isolates. Every child under 16 years of age admitted with NTS in blood culture was approached for recruitment to the study. Malawian replication control samples ($n = 339$) were collected from healthy children at the Ndirande Health Centre, Blantyre, Malawi between October and December 2005, and September and November 2006[34]. Unrelated, healthy children under 16 years of age were eligible for inclusion in the study. Following explanation of the study, written informed consent was obtained from the parent or guardian of each child included in the study, and a buccal swab was taken for DNA extraction. Ethical approval for the study was granted by the College of Medicine Research and Ethics Committee, College of Medicine, University of Malawi.

**NTS discovery GWAS.** Genomic DNA was extracted using QIAamp DNA blood mini kits (Qiagen). Extracted genomic DNA was whole-genome amplified with GenomiPhi (GE Healthcare) according to the manufacturer's instructions. Before whole-genome genotyping, DNA quality was assessed; DNA concentration was assayed using PicoGreen (Invitrogen), agarose gel electrophoresis was used to detect DNA degradation and QC genotypes were assayed using the Sequenom iPLEX assay[35]. Genome-wide genotyping of Kenyan discovery cases and controls was performed with an Affymetrix SNP 6.0 chip. SNP QC filters were applied as follows: minor allele frequency (MAF) <1%, info <0.975, Hardy–Weinberg equilibrium $P < 1 \times 10^{-20}$, plate effect $P < 1 \times 10^{-6}$ and SNP missingness >2%. Following SNP and sample QC (Supplementary Tables 1 & 2) 787,861 autosomal SNPs from 1536 bacteraemia (including 180 NTS) case and 2677 control individuals were taken forward for genome-wide imputation with SHAPEIT[36] and IMPUTE2[37] using all available populations in 1000G Phase1 as a reference panel.

The first four principal components of the genome-wide genotyping data differentiate self-reported ethnicity (Supplementary Fig. 2) and were included in the NTS association analysis to control for underlying population structure. Association analysis was performed using the frequentist score test (accounting for imputed genotype uncertainty) in SNPTEST2[38] under additive and genotypic models of association. Following association analysis, genotyped SNPs nominally associated ($P < 1 \times 10^{-3}$) with NTS were inspected with Evoker and SNPs with poor cluster separation excluded from further analysis. Further SNP QC exclusions were applied as follows: model-specific imputation info score <0.8 and Hardy–Weinberg equilibrium $P < 1 \times 10^{-10}$. Given the study sample size and limited power to replicate associations at less common SNPs (Supplementary Fig. 10), we further excluded SNPs with MAF <10% from the analysis.

**NTS GWAS replication.** SNPs in 16 genomic regions putatively associated with NTS bacteraemia ($P < 1 \times 10^{-6}$) in the Kenyan discovery samples were genotyped in the Kenyan and Malawian replication samples using the Sequenom MASSArray platform. Inspection of cluster plots (TYPER4.0) revealed good cluster separation at 25 SNPs, all of which having a call rate in excess of 95%. Samples were excluded with call rates <80%. Following QC, 416 Malawian samples (135 cases, 281 controls) and 318 Kenyan samples (36 cases, 282 controls) were included in the replication analysis. The Sequenom MASSArray platform was also used to confirm imputation accuracy in 930 Kenyan discovery samples (180 cases, 750 controls) at loci included in the replication analysis (Supplementary Table 4).

The multiplexed Sequenom MASSArray genotyping reaction chosen for replication did not include the STAT4 region SNP (rs13390936) most significantly associated with NTS in the Kenyan discovery samples. This SNP was therefore genotyped by HRMA, using the following PCR primers and an unlabelled probe with a 3′-amino-C7 modification: TAGTGAGCCCTAATGTAAATTATGGGAC (F), CCCTCACCAGTTTCTCCTATATCT (R), GTGATGTACTTGTTACAAATTTATATTATTACAATA (probe). Ten microlitres of PCR reactions (1 μl 10x PCR buffer, 1 μl 25 mM MgCl₂, 1 μl dNTPs (2 mM each), 0.05 μl forward primer (10 μM), 0.025 μl reverse primer (100 μM), 0.025 μl unlabelled probe (100 μM), 1 μl LC Green dye, 0.06 μl HotStarTaq DNA polymerase (Qiagen), 5 μl water and 0.5 μl template DNA (25 ng/μl)) were cycled (95 °C for 10 min, 55 cycles of 30 s at 95 °C, 56 °C and 72 °C), under mineral oil. Following the PCR reaction, the DNA was again denatured (95 °C for 1 min) and cooled to room temperature for 30 min. High-resolution melting was then performed on a LightScanner (Idaho Technology) at 0.1 °C/s increments between 45 °C and 98 °C. Melt curves were analysed with LightScanner Call-IT 2.0 (Idaho Technology) using derivative normalized melting plots between 62 °C and 71 °C (Supplementary Fig. 11). HRMA genotyping of rs13390936 was performed in 1374 Kenyan replication samples (38 cases, 1336 controls) and 489 Malawian samples (150 cases, 339 controls). HRMA genotyping of rs13390936 was also performed in

all Kenyan discovery samples passing sample QC, including six case samples which passed sample QC but failed Affymetrix genotyping (186 cases, 2677 controls).

Kenyan replication cases and controls were also genotyped using the ImmunoChip[39]. Sample QC filters (Supplementary Table 1) were applied as for the discovery samples, resulting in 38 NTS cases and 1336 controls being included in the analysis. These genotypes were used to generate principal components for the replication samples, to correct for population stratification. A LD-pruned set of SNPs ($n = 27,026$) with MAF >1%, call rate >95% (99% if MAF < 5%) and Hardy–Weinberg equilibrium $P < 1 \times 10^{-10}$ were used to generate principal components (Supplementary Fig. 12).

Association analysis of Malawian and Kenyan replication samples was performed in PLINK[40]. For the Kenyan replication samples, the first four principal components of ImmunoChip genotyping data were included in the model to account for population structure (Supplementary Fig. 12). We considered a two-tailed $P$ value <0.05, with the same risk allele in the same model of association as observed in the Kenyan discovery set, as evidence of replication. For the combined analysis, a fixed-effects meta-analysis of the three sample collections was performed in PLINK. To assess evidence for interaction between rs13390936 genotype and HIV, malaria and malnutrition, logistic regression models of NTS disease susceptibility, with and without multiplicative interaction terms, were fitted in R. $P$ values for interactions are calculated with likelihood ratio tests.

**Re-imputation and fine-mapping of the STAT4 region.** A 1 Mb region centred at rs13390936 was intensively re-imputed in the Kenyan discovery samples without pre-phasing with IMPUTE2 using 1000G Phase 3 as a reference panel. Imputation was performed with 1 Mb buffer regions and template haplotypes for phasing increased to 200. This region was also imputed in the Kenyan replication samples from ImmunoChip genotyping data using the same IMPUTE2 settings. Following QC as for the Kenyan discovery samples, association analysis of the re-imputed genotypes was performed using the frequentist score methods in SNPTEST2 under a recessive model of association. Re-imputed genotypes were further used to test for association at STAT4, under a recessive model of association, using a linear mixed model to account for population structure and sample relatedness in GEMMA[41]. In this analysis, samples were not excluded according to pairwise relatedness, but QC thresholds were otherwise as described for the Kenyan discovery samples.

We used a Bayesian approach to identify a set of SNPs with 95% probability of containing the causal locus at the NTS-associated region[42]. Approximate Bayes' factors[43] were calculated under a recessive model for each SNP in the NTS-associated region with a prior distribution of N(0,1). All SNPs were considered equally likely to be the causal variant a priori. A set of SNPs with 95% probability of containing the causal SNP was defined as the smallest number of SNPs for which the summed posterior probabilities exceed 0.95.

**RNA expression quantitative trait analysis of rs13390936.** Using RNA expression and genome-wide genotyping data from published[14–16] and unpublished eQTL studies of naïve and stimulated primary immune cell subsets in healthy European adults, we correlated rs13390936 genotype with STAT4 RNA expression. CD19⁺ B cells, CD14⁺ monocytes and CD56⁺CD3⁻ NK cells were separated from peripheral blood mononuclear cells by magnetic activating cell sorting (Miltenyi). CD16⁺ neutrophils were isolated from granulocytes with CD16⁺ microbeads. Gene expression was quantified in total RNA from naïve cells, and monocytes stimulated with lipopolysaccharide (for 2 or 24 h) and IFNγ (for 24 h), with the Illumina HumanHT-12 v4 BeadChip gene expression array platform. Genome-wide genotyping was performed with the Illumina HumanOmniExpress-12v1.0 Beadchip and imputation undertaken using 1000G phase1 as the reference panel.

Following QC of gene expression and genotyping data as previously described[14–16], rs13390936 genotype was correlated with STAT4 RNA expression in each cell type: B cells ($n = 279$), NK cells ($n = 245$), neutrophils ($n = 101$), naïve monocytes ($n = 414$) and stimulated monocytes (LPS 2 h, $n = 261$; LPS 24 h, $n = 322$; IFNγ 24 h, $n = 367$). Random-spline normalized and variance-stabilization transformed STAT4 expression was correlated with genotype under by linear regression and analysis of variance (ANOVA), including the first 20 principal components of gene expression data in each cell type/condition to account for confounding variation. $P$ values are calculated with $F$-tests (1 d.f.). Statistical analysis was performed in R.

**LD at STAT4 in study populations.** LD plots of the STAT4 region in the Kenyan GWAS discovery samples ($n = 2857$) and the European samples used in the RNA eQTL analyses ($n = 421$) were constructed in Haploview v4.2[44]. To quantify differential LD between the Kenyan and European samples we used varLD[45], quantifying LD variation between the two populations in 50-SNP windows. Raw varLD scores across all of chromosome 2 were standardized, and regions with standardized varLD scores >95th centile (standardized varLD score >1.86) considered to have evidence of differential LD between the two populations.

**Protein phenotypes for rs13390936 in immune cell subsets.** In archived genomic DNA from 5911 individuals registered to the Oxford Biobank (NIHR Oxford Biomedical Research Centre), rs13390936 was genotyped by the Oxford

Biobank laboratory using a TaqMan assay (Applied Biosystems). Fifty-four volunteers (32 females; median age 44 years, range 30–51 years) were recruited to the study according to rs13390936 genotype (36 with the NTS-protective AA genotype, 18 with the NTS-susceptible TT genotype). The study was approved by the Oxfordshire Research Ethics Committee (COREC reference 06/Q1605/55), and written informed consent was obtained from each volunteer.

Whole blood from each volunteer was collected into Lithium Heparin-containing collection tubes (Vacutainer System, Becton Dickinson). One millilitre of aliquots of whole blood from each volunteer were left unstimulated, or stimulated with 10 ng/ml recombinant human IL-12 (catalogue #219-IL, R&D Systems) or $10^6$ cfu/ml NTS (Malawian clinical S. Typhimurium isolate D23580) grown to mid-log phase, within 2 h of collection. In addition, 1 ml whole blood aliquots were stimulated with phorbol 12-myristate 13-acetate (100 ng/ml)/ionomycin (1 μg/ml, Sigma) to act as a positive control. Following stimulation, samples were incubated at 37 °C. Following ex vivo stimulation of each sample, intracellular IFNγ staining was performed. Brefeldin A (Becton Dickinson) was added at 2 h post-stimulation to a final concentration of 2 μg/ml. At 6 h post-stimulation, samples were surface immunostained (20 min, 4 °C) following erythrocyte lysis (Versalyse, Beckman Coulter) with allophycocyanin-conjugated anti-CD3 (clone UCHT1, BD Biosciences), fluorescein isothiocyanate-conjugated anti-CD4 (clone RPA-T4, BD Biosciences) and phycoerythrin (PE)-conjugated anti-CD56 (clone CMSSB, eBioscience) monoclonals. Samples were fixed with Fixation Medium A (Invitrogen), before permeabilization with Permeabilization Medium B (Invitrogen) and intracellular staining with PE-Cy7-conjugated anti-IFNγ monoclonal (clone B27, BD Biosciences) for 20 min at 4 °C.

Immunostained samples were acquired with a BD FACSCanto II and BD FACSDiVa (BD Biosciences), and the data were analysed using FlowJo v10 (TreeStar). To minimize variation across batches, all antibodies were obtained prior to volunteer recruitment and belonged to the same manufacturing lot number. Prior to analysis, CS&T beads of the same lot were used to calibrate the flow cytometer, to ensure equivalent photomultiplier tube voltages between experiments. Proportions of IFNγ+ NK cells were logit transformed prior to analysis, and genotype–phenotype associations tested with linear regression and ANOVA, adjusted for age and sex. P values are calculated with F-tests (1 d.f.). Transformation did not normalize IFNγ+ CD4+ T cell proportions, and genotype–phenotype associations were tested with nonparametric Wilcoxon's rank-sum tests. There was no evidence of significant batch effect ($P < 0.05$) on IFNγ+ cell proportion for either NK cells (linear regression model, adjusted for rs13390936 genotype and stimulation conditions) or CD4+ T cells (Kruskal–Wallis test). There was no evidence of significant inter-genotype inequality of variances (Bartlett test; $P < 0.05$) of IFNγ+ NK cell proportions. Statistical analysis was performed in R.

**Protein phenotypes of rs13390936 in acute NTS bacteraemia**. We collected serum samples from a subset of the Malawian replication cases ($n = 106$). Methods used for bacterial culture and NTS identification and serotyping are as described above. Following identification of NTS in blood culture, the parent or guardian of the child was approached for recruitment to the study. Having obtained written informed consent, a venous blood sample was taken from the child during their acute admission with NTS bacteraemia (median 1 day following collection of the NTS-positive blood culture, range 0–5 days). Plasmodium falciparum parasitaemia was tested with thick and thin Giemsa-stained blood films. HIV status was determined using Determine (Abbot Laboratories) and UniGold (Trinity Biotech) rapid tests, and HIV infection was confirmed in children <18 months by PCR. Children with weight-for-age Z-scores >3 standard deviations below WHO median values were classified as being malnourished. rs13390936 genotypes were assayed as for the Malawian GWAS replication samples (above). Ethical approval for the study was granted by the College of Medicine Research and Ethics Committee, College of Medicine, University of Malawi.

Sera were separated from clotted whole blood within 2 h of venesection, and stored at −80 °C prior to analysis. Serum IFNγ concentrations were assayed using a Bio-Plex Pro 27-plex human cytokine fluorescent bead-based assay (Bio-Rad Laboratories) according to the manufacturer's instructions on a Luminex-100 instrument (Bio-Rad Laboratories) using Bio-Plex Manager 4.1.1 software (Bio-Rad). Assays were performed in three batches with rs13390936 genotypes randomized across batches. IFNγ measurements below the detection limit of the assay were assigned values of the half of the lower detection limit, and were included in the analysis. Correlation of rs13390936 genotype with serum IFNγ concentration during acute NTS bacteraemia was performed by linear regression and ANOVA, adjusting for age, sex and NTS-associated co-morbidities (HIV, malnutrition and malaria). Serum IFNγ concentrations were cube-root transformed prior to analysis. P values are calculated with F-tests (1 d.f.). There was no evidence of significant inter-genotype inequality of variances (Bartlett test; $P < 0.05$) of IFNγ concentration. Statistical analysis was performed in R.

**Model comparisons of rs13390936 in children with bacteraemia**. We compared models of association at rs13390936 across the six most frequently isolated bacterial pathogens among cases of bacteraemia in the Kenyan discovery samples in the all-cause bacteraemia GWAS[10]. Two thousand six hundred and fifty-one control samples and 1222 cases (Streptococcus pneumoniae, 427 cases; NTS, 177 cases; Staphylococcus aureus, 172 cases; Escherichia coli, 157 cases; β-haemolytic

Streptococci, 156 cases; Haemophilus influenzae type b, 133 cases) were included in the analysis. Effect size estimates and 95% CIs were calculated by multinomial logistic regression under a recessive model, using control status and each of the bacterial pathogen subgroups as strata, and the first 4 principal components as covariates (Fig. 3a). We considered four models of effect across the bacterial pathogens, defined by the prior distributions on the effect size:

NULL : effect size = 0, i.e. no association with any pathogen
SAME: effect size ∼ N(0, 1) and fixed between pathogens ($\rho = 1$)
REL: effect size ∼ N(0, 1) and correlated ($\rho = 0.96$),
but not fixed, between pathogens
NTS : effect size ∼ N(0, 1) for NTS and is zero for all other pathogens

For each model we calculated approximate Bayes factors[43] and posterior probabilities (Fig. 3b), assuming each model to be equally likely a priori. Statistical analysis was performed in R.

Using the same approach, we compared models of association at rs13390936 in subgroups of children with NTS bacteraemia in the Kenyan discovery samples. We compared models of association in children with NTS bacteraemia secondary to S. Enteritidis and S. Typhimurium (82 S. Enteritidis cases, 65 S. Typhimurium cases, 2651 controls), and in children with NTS bacteraemia with and without acquired risk factors for NTS disease (57 NTS cases with malaria, 129 NTS cases without malaria, 2651 controls; 65 NTS cases with malnutrition, 101 NTS cases without malnutrition, 2651 controls).

**Model comparisons of rs13390936 in autoimmune diseases**. To compare evidence for association at rs13390936 across a range of autoimmune diseases, we downloaded from ImmunoBase[21] summary statistics of disease associations at rs16833239 (in perfect LD with rs13390936 in CEU 1000 Genomes Pilot 1 population; $r^2 = 1$, $D' = 1$) for nine association studies of eight autoimmune diseases (coeliac disease, rheumatoid arthritis, primary biliary cirrhosis, multiple sclerosis, narcolepsy, juvenile idiopathic arthritis, psoriasis and type 1 diabetes—one case–control study and one meta-analysis of case–control and family-based studies) conducted using the ImmunoChip[39] in individuals of European ancestry[18–21,46–49]. Standard errors were estimated from reported odds ratios and P values calculated under an additive model (Fig. 3c).

**Consortia**. Sample collection, DNA extraction and phenotyping of the Kenyan children presented in this study were generated by the Kenyan Bacteraemia Study Group. Genome-wide genotyping data of the Kenyan samples was generated by the Wellcome Trust Case Control Consortium 2. Memberships of both consortia are provided in the Supplementary Information.

**Data availability**. Genotype and phenotype data for Kenyan discovery and replication samples are available via the European Genotype Archive, with the accession code EGAS00001001756. Genotype and phenotype data for Malawian replication samples are available on request from the corresponding authors.

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

## Acknowledgements

The principal funding for this study was provided by the Wellcome Trust, as part of the Wellcome Trust Case Control Consortium 2 project (grants 084716/Z/08/Z, 085475/B/ 08/Z and 085475/Z/08/Z). This work was partially supported by Wellcome Trust Centre for Human Genetics core grant 090532/Z/09/Z. The fieldwork and phenotyping in Kenya was supported by the Kenya Medical Research Institute (KEMRI) and the Wellcome Trust of Great Britain. The fieldwork and phenotyping in Malawi was supported by a Wellcome Trust Research Fellowship (067902/Z/02/Z) to C.A.M., and a Wellcome Trust Programme Grant (074124/Z/04/Z) to M.E.M. J.J.G. is supported by a Wellcome Trust Clinical PhD Fellowship (102342/Z/13/Z), A.R. was supported by the Wellcome Trust (084716/Z/08/Z) and the European Research Council, T.N.W. and J.A.G.S. were supported by Senior Research Fellowships from the Wellcome Trust (091758 and 098532 respectively), S.J.C. was supported by the NIHR Biomedical Research Centre, Oxford, C. C.A.S. was supported by a Wellcome Trust Career Development Fellowship (097364/Z/ 11/Z) and A.V.S.H. is supported by a Wellcome Trust Senior Investigator Award (HCUZZ0) and by a European Research Council advanced grant (294557). This study makes use of data generated by MalariaGEN. A full list of the investigators who contributed to the generation of the data is available from www.MalariaGEN.net. Funding for this project was provided by Wellcome Trust (WT077383/Z/05/Z) and the Bill & Melinda Gates Foundation through the Foundation of the National Institutes of Health (566) as part of the Grand Challenges in Global Health Initiative. We thank all the study participants and Kilifi District Hospital and Queen Elizabeth Central Hospital clinical teams and laboratory staff for their involvement in data and sample collection. We thank the volunteers from the Oxford Biobank, NIHR Oxford Biomedical Research Centre, for their participation. The Oxford Biobank (www.oxfordbiobank.org.uk) is also part of the NIHR National Bioresource which supported the recalling process of the volunteers. We thank the High-Throughput Genomics Group at the Wellcome Trust Centre for Human Genetics (funded by Wellcome Trust grant reference 090532/Z/09/Z and MRC Hub grant G0900747 91070) for the generation of the Sequenom data. This paper was published with the permission of the Director of KEMRI.

## Author contributions

Author contributions were as follows: J.J.G., A.R, H.T., M.P., and C.C.A.S. performed the statistical and computational analysis. J.A.G.S., T.N.W., E.M., S.M., P.N., N.M. and The Kilifi Bacteraemia Surveillance Group recruited Kenyan study subjects and compiled phenotypic data. C.A.M., C.L.M., E.N.G., J.M.M, and M.E.M. recruited Malawian study subjects and compiled phenotypic data. DNA extraction, sample handling and genotyping was performed by J.J.G, A.R, T.C.M., C.L.M., The Kilifi DNA extraction Group and The Wellcome Trust Case Control Consortium 2, DNA, Genotyping, Data QC and Informatics Group. J.J.G., V.N., B.P.F., J.C.K. and A.V.S.H. designed and performed the Oxford Biobank study. J.J.G., A.R. and A.V.S.H. wrote the manuscript. The study was designed and managed by J.J.G., A.R., B.P.F., V.N., J.C.K, S.J.C., C.A.M., T.N.W., J.A.G.S., A.V.S.H. and the Wellcome Trust Case Control Consortium 2.

## Additional information

**Competing interests:** The authors declare no competing financial interests.

