## [Peer Review File · Nature Communications]

Reviewers' comments:

Reviewer #1 (Remarks to the Author):

This study identified a non-coding SNP (rs13390936) that can affect STAT4 expression and INF-g level in nontyphoidal Salmonella (NTS) patients using genome-wide association study (GWAS). Although the sample size of 180 cases is relatively small, the presence of multiple SNPs having significant p values near the rs13390936 locus and in LD further supports the association. Additionally, the data from in vitro stimulation study and in vivo serum IFN-g levels in the NTS patients further support the conclusion. The authors also showed that the association is specific for NTS, not for other selected infections. The risk allele (TT) of the STAT4 was shown to be protective for some autoimmune diseases.

The data are well presented and are convincing in supporting the conclusion. The only negative part is that the story of STAT4 mutation—change in IFN-g expression—variation in infection protection /autoimmune risk is somehow expected because the signaling pathways and the role of IFN-g in inflammation and protection in infections are also known. Still this is an interesting and important study.

Specific minor points"

1. There are some numbers that are confusing 930 Kenyan (line 86), 1374 Kenyan (line 89), and 180 case/2677 control Kenyan (line 682). Please clarify the relationships of these numbers.
2. Line 194, "modulating IFN-g production capacity in NK cells, but not CD4+ cells". Why, please discuss.
3. The font sizes in figure 2 and 3 are small, difficult to read.

Reviewer #2 (Remarks to the Author):

The authors performed GWAS for nontyphoidal Salmonella (NTS) bacteraemia in African children and identified genome-wide significant association of an intronic variant of STAT4 gene. The finding was further supported by functional studies: decreased STAT4 expression in LPS- or IFN γ - stimulated NK cells and decreased serum IFN γ levels during acute NTS bacteraemia. Their approaches and results are clearly described. However, there are several points to be clarified.

Major points

- 1) The authors mentioned that HIV infection, malnutrition, and malaria contribute to susceptibility to invasive NTS disease, but these factors were not considered in the present GWAS. Since there are differences in proportions of HIV and malaria infection (Supplementary Table VII) and in the observed odds ratios (Table 1) between Kenyan and Malawian groups, possible contribution of these factors to the risk of STAT4 variant should be evaluated.
- 2) The authors performed GWAS under both genotypic and additive models and there is difference in the number of SNPs assessed in each model (5,585,198 SNPs and 4,669,480 SNPs, respectively). There is no description about the reason of the difference. Information of the number of SNPs excluded in each SNP QC step is helpful.
- 3) Considering the limited number of replication samples, I agree to focus on the SNPs with MAF > 10%. However, there is no description about this limitation in the text.

- 4) To assess the confound effect of HIV-infection, the association of rs13390936 genotype with IFN γ production among HIV-infected Malawian children should also be discussed.
- 5) Table I and Supplementary Table IV: There is inconsistency for the p-value and OR of rs13390936.
- 6) The authors may discuss the reason why eQTL was observed in NK cells but not in CD4+ T cells, and should discuss the importance of NK cells in NTS infection.
- 7) Table 1: The numbers of individuals with TT genotype were small: just 3 for Kenyan replication cases and Malawian replication controls. Did the authors applied Fisher's exact test for the p-value under recessive model?

Minor points:

- 1) For numbering of Tables, Roman numerals are used.
- 2) Supplementary Table II: As the authors used three different methods for genotyping, it is not clear how many samples were typed by each method.
- 3) Supplementary Figure 1: Their final conclusion on the genetic model was 'recessive model'. The inflation factor under the recessive model should be presented.
- 4) Methods: Regarding the genome-wide imputation, it is unclear whether the all population data or specific population data were used. The information should be mentioned for each imputation analysis.
- 5) Table 1, Supplementary Table II: They just mentioned as "combined analysis". I guess the analysis is actually "meta-analysis" as mentioned for Supplementary Table IV.

Reviewer #3 (Remarks to the Author):

1. The general concern with the presented study is the small sample sizes of the discovery and replication groups. Genetic studies with small sample sizes are prone to produce more significant results and more false positive ones. Although functional evidence was presented that the identified SNP might alter STAT4 expression I'm not convinced that the genetic association analysis was done properly as a number of correction factors were not adjusted for. In the present form the manuscript is not helpful clarifying the question whether human genetic variants predispose to NTS infections.
2. HIV infection, malnutrition and malaria are factors predisposing for NTS infection. However, no corrections for the occurrences of those factors were performed neither in the discovery nor in the replication group. In addition, not all individuals of the study groups were tested for the presence of HIV and malnutrition as shown in supplementary table VII. Of 180 included participants of the Kenyan study group only 117 appears to be tested for HIV and 160 for wasting. Unrecognized viral infections might introduce a bias and all study members should have been tested for the viral infection. Furthermore, as the occurrence of severe malaria is influenced by the sickle cell allele HBS allele, the influence of HBS on NTS should also be analysed.
3. NTS serovars were determined in all study groups but no analyses were presented whether the presence of one or the other strain might influence the association study. The proportions of Typhimurium and Enteritidis differ extremely between the Kenyan and Malawian groups (39.8% vs 50.6%

Kenya and 90.0% vs 8.0% Malawi). An analysis should be included whether the different strains and the discrepancy between the Kenyan and Malawian studies influence the outcome of the genetic association study.

4. Different thresholds for exclusion of SNPs due to deviations from HWE in the Kenyan group were given ($P < 1 \times 10^{-10}$ and $P < 1 \times 10^{-20}$). The authors should explain the discrepancies and also the extreme low threshold of $P < 1 \times 10^{-20}$.

5. The genotypic model for testing for difference between genotypes were given which only is a rough measure for discrepant genotype distributions and is rather unusual in genetic association testing as no direction of effect could

6. Exclusion of participants due to relatedness with $IBD > 0.4$ is not very strict and lies between the IBD value for full sibs with 0.5 and half-sibs of 0.25. A threshold of 0.25 would therefore be more reliable and minimize the risk of including related individuals in particular when participants were enrolled in the same villages.

Reviewer Responses

Risk of nontyphoidal *Salmonella* bacteraemia in African children is modified by *STAT4*

Reviewer #1 (Remarks to the Author):

This study identified a non-coding SNP (rs13390936) that can affect *STAT4* expression and INF-g level in nontyphoidal *Salmonella* (NTS) patients using genome-wide association study (GWAS). Although the sample size of 180 cases is relatively small, the presence of multiple SNPs having significant p values near the rs13390936 locus and in LD further supports the association. Additionally, the data from in vitro stimulation study and in vivo serum INF-g levels in the NTS patients further support the conclusion. The authors also showed that the association is specific for NTS, not for other selected infections. The risk allele (TT) of the *STAT4* was shown to be protective for some autoimmune diseases.

The data are well presented and are convincing in supporting the conclusion. The only negative part is that the story of *STAT4* mutation—change in INF-g expression—variation in infection protection /autoimmune risk is some how expected because the signaling pathways and the role of INF-g in inflammation and protection in infections are also known. Still this is an interesting and important study.

Specific minor points

1. There are some numbers that are confusing 930 Kenyan (line 86), 1374 Kenyan (line 89), and 180 case/2677 control Kenyan (line 682). Please clarify the relationships of these numbers.

Many thanks for highlighting that this was not adequately clear. 180 case/2,677 controls are the total Kenyan discovery numbers post-quality control (QC), and 38 case/1,336 controls (total 1,374) are total Kenyan replication numbers post-QC. The 930 Kenyan samples were a subset of the post-QC Kenyan discovery samples used to confirm imputation accuracy. To make it clearer that the 930 Kenyan samples are a subset of the discovery samples, we have amended the following sentence to now read (changes highlighted):

“Imputation accuracy of SNPs taken forward for replication genotyping was confirmed in 930 of the Kenyan discovery samples (33% of the discovery samples; 180 cases and 750 controls) by Sequenom MASSArray (mean $r^2=0.97$, Supplementary Table 4).”

To clarify that the 1,374 Kenyan samples included in the replication are separate to the samples included in the discovery experiment, we have amended the following sentence to now read (changes highlighted):

“Following the discovery analysis, an additional 1,374 Kenyan samples (38 cases, 1,336 controls) and 489 Malawian samples (150 cases, 339 controls) were included in the replication genotyping (Supplementary Table 4).”

2. Line 194, “modulating INF-g production capacity in NK cells, but not CD4+ cells”. Why, please discuss.

Many thanks for this suggestion. We have added the following discussion to the main text of the article:

“The identification of genetic variation modifying INF γ production in NK cells, but not CD4+ T cells, as a risk factor for NTS bacteraemia is noteworthy. These observations are in keeping with findings in the mouse model of NTS infection, in which NK cells have been

shown to have a protective role in anti-Salmonella immunity, in an IFN γ -dependent manner²⁹. The observed association between NTS-associated genetic variation, NK cell IFN γ production, and serum IFN γ production in acute NTS bacteraemia, suggests that NK cells are an important source of IFN γ in NTS infections in African children. The lack of a similar association between NTS-associated genetic variation and CD4⁺ T cell IFN γ production does not preclude an important role for CD4⁺ T cell-derived IFN γ in anti-Salmonella immunity in African children. It is, however, consistent with the observation that acquisition of Salmonella-specific CD4⁺ T cells in Malawian children does not result in effective anti-Salmonella immunity³⁰."

3. The font sizes in figure 2 and 3 are small, difficult to read.

Very many thanks for your comments. Font sizes in Figures 2 and 3 have been amended accordingly.

Reviewer #2 (Remarks to the Author):

The authors performed GWAS for nontyphoidal Salmonella (NTS) bacteraemia in African children and identified genome-wide significant association of an intronic variant of STAT4 gene. The finding was further supported by functional studies: decreased STAT4 expression in LPS- or IFN γ -stimulated NK cells and decreased serum IFN γ levels during acute NTS bacteraemia. Their approaches and results are clearly described. However, there are several points to be clarified.

Major points

1) The authors mentioned that HIV infection, malnutrition, and malaria contribute to susceptibility to invasive NTS disease, but these factors were not considered in the present GWAS. Since there are differences in proportions of HIV and malaria infection (Supplementary Table VII) and in the observed odds ratios (Table 1) between Kenyan and Malawian groups, possible contribution of these factors to the risk of STAT4 variant should be evaluated.

Many thanks for raising this important point. To address this, we have included additional analysis of the Malawian replication dataset, in which we include HIV co-infection, severe malnutrition and malaria as covariates in a model of association between rs13390936 and NTS disease. In that model, the observed association between rs13390936 genotype and NTS disease is unchanged. We further explore the possibility of multiplicative interactions between rs13390936 genotype and HIV, severe malnutrition and malaria, identifying no evidence for interaction between rs13390936 and malaria, HIV or malnutrition. We have added the following to the manuscript to reflect these changes:

"Established NTS risk factors and rs13390936

Malaria, HIV infection and malnutrition are important, acquired risk factors for NTS bacteraemia in African children^{6,7}. To address whether the observed association of rs13390936 with NTS bacteraemia is independent of these acquired risk factors, we fitted a regression model of rs13390936 association with NTS bacteraemia in the Malawian replication samples, including HIV infection, malaria and severe malnutrition as covariates. In that model, including 396 Malawian children (109 cases, 287 controls) for which complete covariate data is available, rs13390936 is associated with NTS bacteraemia under a recessive model ($P=9.96 \times 10^{-3}$; OR 7.74, 95% CI, 6.18-9.30) independent of HIV, malaria and malnutrition. Rates of HIV and malaria co-infection, but

not malnutrition, among Kenyan and Malawian children with NTS included in the study are significantly different (Supplementary Table 7). The observed effect size of the rs13390936 association with NTS bacteraemia is lower in Malawian children than that observed in Kenyan children, albeit with no evidence of inter-study heterogeneity of effect (Table 1). To investigate whether there is evidence for interaction between rs13390936 genotype and HIV, malaria and malnutrition, we fitted regression models including interaction terms between rs13390936 genotype and each acquired risk factor, in the same set of Malawian children. In that analysis, there is no evidence for interaction between carriage of the rs13390936:TT genotype and HIV status ($P=0.81$), malaria ($P=0.07$) or malnutrition ($P=0.82$)."

In addition, we have added the following to the methods section:

"To assess evidence for interaction between rs13390936 genotype and HIV, malaria and malnutrition, logistic regression models, with and without multiplicative interaction terms, were fitted in R. P-values for interactions are calculated with likelihood ratio tests."

We are only able to perform this analysis in the Malawian samples. The Kenyan control samples are a birth cohort, and we could not therefore account for transient acquired risk factors (e.g. malaria, malnutrition) in those samples, and HIV status is not available for the Kenyan control samples. We now highlight this limitation in the discussion, as follows: *"A consequence of our use of a birth cohort as the control population for the Kenyan samples in this study is that we were unable to correct for transient risk factors (e.g. malaria and malnutrition) in the discovery analysis. We demonstrate that risk of NTS bacteraemia conferred by rs13390936 genotype is independent of malaria, malnutrition and HIV status in Malawian children. However, it will be important to establish in future studies, whether the association between rs13390936 and risk of NTS bacteraemia is observed consistently across populations of African children with diverse rates of HIV, malaria and malnutrition."*

2) The authors performed GWAS under both genotypic and additive models and there is difference in the number of SNPs assessed in each model (5,585,198 SNPs and 4,669,480 SNPs, respectively). There is no description about the reason of the difference. Information of the number of SNPs excluded in each SNP QC step is helpful.

We thank Review #2 for highlighting this. The discrepancy between the number of SNPs included in the additive and genotypic analysis is due to the use of model-specific imputation info scores in the post-imputation QC process. As suggested the number of SNPs removed at each QC step (pre- and post-imputation) is now stated in a new supplementary table (Supplementary Table 2):

Genotyped SNP exclusions	Minor allele frequency <1%	50,322	
	Info <0.975	53,419	
	HWE $P < 1 \times 10^{-20}$	18,288	
	Plate effect $P < 1 \times 10^{-6}$	7,382	
	SNP missingness >2%	34,430	
Genotyped, autosomal SNPs taken forward for imputation		787,861	
		Additive model	Genotypic model
Imputed SNP exclusions	HWE $P < 1 \times 10^{-10}$	1,684	
	Minor allele frequency <10%	8,774,454	
	Imputation info <0.8*	572,862	1,439,881
	Imputed, autosomal SNPs included in analysis	5,585,198	4,669,480

Supplementary Information Table 2 | Kenyan discovery GWAS SNP exclusions. SNP exclusions in the Affymetrix SNP 6.0 chip-genotyped Kenyan discovery samples, pre- and post-imputation. HWE, Hardy Weinberg Equilibrium.

*While the imputation info threshold (0.8) is used in both additive and genotypic analyses, the metric is model specific, resulting in larger numbers of SNP exclusions in the genotypic analysis.

3) Considering the limited number of replication samples, I agree to focus on the SNPs with MAF > 10%. However, there is no description about this limitation in the text.

We agree that this should be highlighted in the main text. We have added the following to the discussion:

“The limited sample size, in particular with respect to case numbers, restricted this analysis to the contribution of common genetic variation (minor allele frequency >10%) to risk of NTS bacteraemia. Larger sample sizes will be required to assess the contribution of less common genetic variants, and variants with smaller effect sizes, on the risk of NTS bacteraemia in African children.”

4) To assess the confound effect of HIV-infection, the association of rs13390936 genotype with IFN γ production among HIV-infected Malawian children should also be discussed.

Many thanks for this suggestion. We now present the association of rs13390936 genotype with serum IFN levels in both HIV-infected and –uninfected children in Supplementary Figure 7.

We have added the following to the manuscript main text:

“In HIV-infected children (n=43), there is no evidence for association between serum IFN γ levels during acute NTS bacteraemia and rs13390936 genotype ($P = 0.23$, Supplementary Fig. 7). The lack of association between rs13390936 genotype and serum IFN γ may simply reflect a lack power in HIV-infected children, or may reflect other sources of variation in IFN γ response in HIV-infected children (e.g. CD4 $^+$ T cell count).”

5) Table I and Supplementary Table IV: There is inconsistency for the p-value and OR of rs13390936.

Many thanks for highlighting this. The discrepancy between Table 1 and Supplementary Table IV (now Supplementary Table 6) reflect the difference between the estimated effect sizes using directly genotyped genotypes (Table 1) and imputed genotypes (Supplementary Table IV, now 6). To make this clearer, we have added the following to figure legend for Supplementary Table IV (now Supplementary Table 6):

“Discrepancies between p-values and effect sizes displayed for Kenyan discovery samples displayed here, and those reported in Table 1, reflect the use of imputed genotypes (here) and directly assayed genotypes (Table 1).”

6) The authors may discuss the reason why eQTL was observed in NK cells but not in CD4⁺ T cells, and should discuss the importance of NK cells in NTS infection.

Many thanks for making this suggestion, along with Reviewer #1. We have added the following discussion to the main text of the article:

“The identification of genetic variation modifying IFN γ production in NK cells, but not CD4⁺ T cells, as a risk factor for NTS bacteraemia is noteworthy. These observations are in keeping with findings in the mouse model of NTS infection, in which NK cells have been shown to have a protective role in anti-Salmonella immunity, in an IFN γ -dependent manner²⁹. The observed association between NTS-associated genetic variation, NK cell IFN γ production, and serum IFN γ production in acute NTS bacteraemia, suggests that NK cells are an important source of IFN γ in NTS infections in African children. The lack of a similar association between NTS-associated genetic variation and CD4⁺ T cell IFN γ production does not preclude an important role for CD4⁺ T cell-derived IFN γ in anti-Salmonella immunity in African children. It is, however, consistent with the observation that acquisition of Salmonella-specific CD4⁺ T cells in Malawian children does not result in effective anti-Salmonella immunity³⁰.”

7) Table 1: The numbers of individuals with TT genotype were small: just 3 for Kenyan replication cases and Malawian replication controls. Did the authors applied Fisher’s exact test for the p-value under recessive model?

Many thanks for this point. Effect estimates for Kenyan and Malawian replication controls were calculated using logistic regression. Using Fisher’s exact tests would not have allowed for the inclusion of covariates (e.g. to account for population structure) in these models. We now provide Fisher’s exact estimates of effect size in the Kenyan and Malawian replication samples in the Table 1 legend as follows:

“Fisher exact estimates of effect sizes in the Kenyan (OR=9.56, P=6.79x10⁻³) and Malawian (OR=4.87, P=0.023) replication samples (without covariates) are comparable to those derived by logistic regression.”

Minor points:

1) For numbering of Tables, Roman numerals are used.

Many thanks, this has been corrected throughout.

2) Supplementary Table 2: As the authors used three different methods for genotyping, it is not clear how many samples were typed by each method.

Many thanks for highlighting the need to clarify this. We have amended the Supplementary Table 2 (now Supplementary Table 4) legend to now include:

*“Genotype data was generated by imputation using the Affymetrix SNP 6.0 chip (Kenyan discovery samples) and Sequenom MASSArray (Kenyan replication and Malawian replication). Where ImmunoChip genotyping data was available (SNPs highlighted**), these data are presented. rs13390936 (highlighted in bold) was genotyped by High-Resolution Melt-curve Analysis (HRMA). Sample numbers are as indicated in the table, with the exception of ImmunoChip genotyped SNPs (Kenyan replication, 38 cases, 1,336 controls) and rs13390936 (Kenyan replication, 38 cases, 1,336 controls; Malawian replication, 150 cases, 339 controls).”*

3) Supplementary Figure 1: Their final conclusion on the genetic model was ‘recessive model’. The inflation factor under the recessive model should be presented.

Many thanks for this suggestion. We have added the following to the Supplementary Figure 1 legend:

“For reference, genome-wide inflation factors under recessive dominant and heterozygous advantage models of association are 1.02, 1.00, and 1.00 respectively.”

4) Methods: Regarding the genome-wide imputation, it is unclear whether the all population data or specific population data were used. The information should be mentioned for each imputation analysis.

All imputation was performed using all available reference populations in the 1000 genomes project. We have clarified this in the methods as follows (addition highlighted):

*“Following SNP and sample QC (Supplementary Tables 1 & 2) 787,861 autosomal SNPs from 1,536 bacteraemia (including 180 NTS) case and 2,677 control individuals were taken forward for genome-wide imputation with SHAPEIT³¹ and IMPUTE2³² using **all available populations** in 1000G Phase1 as a reference panel.”*

5) Table 1, Supplementary Table II: They just mentioned as “combined analysis”. I guess the analysis is actually “meta-analysis” as mentioned for Supplementary Table IV.

Many thanks for highlighting this inconsistency. The header on Supplementary Table 2 (now Supplementary Table 4) has been amended accordingly.

Reviewer #3 (Remarks to the Author):

1. The general concern with the presented study is the small sample sizes of the discovery and replication groups. Genetic studies with small sample sizes are prone to produce more significant results and more false positive ones. Although functional evidence was presented that the identified SNP might alter STAT4 expression I’m not convinced that the genetic association analysis was done properly as a number of correction factors were not adjusted for. In the present form the manuscript is not helpful clarifying the question whether human genetic variants predispose to NTS infections. **Many thanks for taking the time to review our work. We respond to your specific concerns, regarding appropriate correction for covariates in the genetic association analysis, below.**

2. HIV infection, malnutrition and malaria are factors predisposing for NTS infection. However, no corrections for the occurrences of those factors were performed neither in the discovery nor in the replication group.

Many thanks for highlighting this important point along with Reviewer #2. As above, we have included additional analysis of the Malawian replication dataset, in which we include HIV co-infection, severe malnutrition and malaria parasitaemia as covariates in a model of association

between rs13390936 and NTS disease. In that model, the observed association between rs13390936 genotype and NTS disease is unchanged. We further explore the possibility of multiplicative interactions between rs13390936 genotype and HIV, severe malnutrition and malaria, identifying no evidence for interaction between rs13390936 and malaria, HIV or malnutrition. We have added the following to the manuscript to reflect these changes:

“Established NTS risk factors and rs13390936

Malaria, HIV infection and malnutrition are important, acquired risk factors for NTS bacteraemia in African children^{6,7}. To address whether the observed association of rs13390936 with NTS bacteraemia is independent of these acquired risk factors, we fitted a regression model of rs13390936 association with NTS bacteraemia in the Malawian replication samples, including HIV infection, malaria and severe malnutrition as covariates. In that model, including 396 Malawian children (109 cases, 287 controls) for which complete covariate data is available, rs13390936 is associated with NTS bacteraemia under a recessive model ($P=9.96 \times 10^{-3}$; OR 7.74, 95% CI, 6.18-9.30) independent of HIV, malaria and malnutrition. Rates of HIV and malaria co-infection, but not malnutrition, among Kenyan and Malawian children with NTS included in the study are significantly different (Supplementary Table 7). The observed effect size of the rs13390936 association with NTS bacteraemia is lower in Malawian children than that observed in Kenyan children, albeit with no evidence of inter-study heterogeneity of effect (Table 1). To investigate whether there is evidence for interaction between rs13390936 genotype and HIV, malaria and malnutrition, we fitted regression models including interaction terms between rs13390936 genotype and each acquired risk factor, in the same set of Malawian children. In that analysis, there is no evidence for interaction between carriage of the rs13390936:TT genotype and HIV status ($P=0.81$), malaria co-infection ($P=0.07$) or malnutrition ($P=0.82$).”

In addition, we have added the following to the methods section:

“To assess evidence for interaction between rs13390936 genotype and HIV, malaria and malnutrition, logistic regression models, with and without multiplicative interaction terms, were fitted in R. P-values for interactions are calculated with likelihood ratio tests.”

We are only able to perform this analysis in the Malawian samples. The Kenyan control samples are a birth cohort, and we could not therefore account for transient acquired risk factors (e.g. malaria, malnutrition) in those samples, and HIV status is not available for the Kenyan control samples. We now highlight this limitation in the discussion, as follows:

“A consequence of our use of a birth cohort as the control population for the Kenyan samples in this study, is that we were unable to correct for transient risk factors (e.g. malaria and malnutrition) in the discovery analysis. We demonstrate that risk of NTS bacteraemia conferred by rs13390936 genotype is independent of malaria, malnutrition and HIV status in Malawian children. However, it will be important to establish in future studies, whether the association between rs13390936 and risk of NTS bacteraemia is observed consistently across populations of African children with diverse rates of HIV, malaria and malnutrition.”

In addition, not all individuals of the study groups were tested for the presence of HIV and malnutrition as shown in supplementary table VII. Of 180 included participants of the Kenyan study group only 117 appears to be tested for HIV and 160 for wasting. Unrecognized viral infections might introduce a bias and all study members should have been tested for the viral infection.

We agree with Reviewer #3 that it would be optimal to have co-morbidity data available for all study participants. That the comorbidity data is incomplete simply reflects the difficulties of recruiting and phenotyping children presenting with invasive bacterial infections with high and early mortality in resource-poor settings. The potential for the introduction of study bias secondary to under-ascertainment of HIV co-infection with respect to the rs13390936 association with NTS seems to us to be remote. As detailed above, in Malawian children we find no evidence that this association is confounded by HIV infection, nor is there any evidence of significant interaction between HIV and rs13390936 genotype. Moreover, among the previously published genome-wide association studies of HIV-associated traits (including susceptibility, viral load set-point and CD4:CD8 ratio), there is no evidence for a genetic association at *STAT4*. To better acknowledge the limited availability of co-morbidity data in our study population, we have added the following to the discussion:

“However, it will be important to establish in future studies, whether the association between rs13390936 and risk of NTS bacteraemia is observed consistently across populations of African children with diverse rates of HIV, malaria and malnutrition.”

Furthermore, as the occurrence of severe malaria is influenced by the sickle cell allele HBS allele, the influence of HBS on NTS should also be analysed.

We thank Reviewer #3 for raising this important point. To address this we now present association of rs334 on NTS bacteraemia (and the risk/protection conferred by HbSS and HbAS carriage) in a new Supplementary Figure 4. In that figure, we also re-analyse the NTS association at the *STAT4* locus, demonstrating that the observed association between NTS bacteraemia and genotype at the *STAT4* locus is independent of genotype (HbAS or HbSS carriage) at the sickle cell locus. We summarise these additional analyses in the main text as follows:

*“The sickle cell locus (rs334) has been previously demonstrated to be associated with NTS bacteraemia in Kenyan children⁵ (in sickle cell disease - HbSS), and with protection against malaria, and as a consequence, bacteraemia⁷ (sickle cell trait - HbAS). In the Kenyan discovery samples (164 cases, 2,342 controls), sickle cell disease is associated with increased risk of NTS bacteraemia ($P=8.30 \times 10^{-5}$; OR 4.89 95% CI 2.10-10.40), and there is no statistically significant evidence ($P=0.65$) for an effect of sickle cell trait (HbAS) on risk of NTS bacteraemia (Supplementary Fig. 4). As carriage of HbSS is associated with risk of NTS bacteremia, and carriage of HbAS is associated with a key risk factor for NTS bacteraemia (malaria), we sought to assess whether the observed association at the *STAT4* locus is independent of genotype at rs334. In regression models conditioned on genotype at rs334 (Supplementary Fig. 4), the observed association at rs13390936 with NTS bacteraemia persists when conditioned on HbAS (OR 7.16 95% CI 2.68-17.41) and HbSS (OR 8.08 95% CI 2.71-21.89).”*

3. NTS serovars were determined in all study groups but no analyses were presented whether the presence of one or the other strain might influence the association study. The proportions of Typhimurium and Enteritidis differ extremely between the Kenyan and Malawian groups (39.8% vs 50.6% Kenya and 90.0% vs 8.0% Malawi). An analysis should be included whether the different strains and the discrepancy between the Kenyan and Malawian studies influence the outcome of the genetic association study.

Many thanks for highlighting this important point. To address this we have performed a Bayesian comparison of models of association at rs13390936 with bacteraemia secondary to *S. Enteritidis* and *S. Typhimurium*. In that analysis the most probable model is one in which the risk conferred by rs13390936 is the same in both *S. Typhimurium* and *S. Enteritidis*

bacteraemia. This analysis is presented in a new Supplementary Figure 9, and is presented in the main text as follows:

“Again using a Bayesian approach to compare models of association at rs13390936, we further investigated whether the increased risk of NTS bacteraemia conferred by rs13390936 genotype is specific to the two major NTS serovars causing disease in African children, S. Typhimurium and S. Enteritidis. In that analysis, in the Kenyan discovery samples, the most probable model is one in which rs13390936 is associated with susceptibility to bacteraemia secondary to both S. Typhimurium and S. Enteritidis (Supplementary Fig. 9). This suggests that, in keeping with these serovars causing clinically indistinguishable syndromes, and possessing the same acquired risk factors¹⁵, that host genetic risk factors for NTS bacteraemia are shared by both of the major causative serovars.”

4. Different thresholds for exclusion of SNPs due to deviations from HWE in the Kenyan group were given ($P < 1 \times 10^{-10}$ and $P < 1 \times 10^{-20}$). The authors should explain the discrepancies and also the extreme low threshold of $P < 1 \times 10^{-20}$.

Many thanks for highlighting that this was not adequately clear. The first HWE threshold applied ($P < 1 \times 10^{-20}$) was applied prior to imputation, to remove poorly genotyped SNPs. This threshold was adopted as common practice among the WTCCC2 studies, and in our analysis resulted in the removal of 18,288 SNPs pre-imputation. The second HWE threshold ($P < 1 \times 10^{-10}$) was applied post-imputation to further remove poorly genotyped SNPs, but also to remove poorly imputed SNPs. To make the SNP QC process clearer, we now present this in a new Supplementary Table 2.

5. The genotypic model for testing for difference between genotypes were given which only is a rough measure for discrepant genotype distributions and is rather unusual in genetic association testing as no direction of effect could

Many thanks for this point. Our use of the genotypic model in this context was motivated by the observation that, in our previously-published analysis of all-cause bacteraemia in this population, the established susceptibility locus for bacteraemia (rs334, the sickle cell locus) was identifiable in a genotypic model, but not an additive model.

6. Exclusion of participants due to relatedness with $IBD > 0.4$ is not very strict and lies between the IBD value for full sibs with 0.5 and half-sibs of 0.25. A threshold of 0.25 would therefore be more reliable and minimize the risk of including related individuals in particular when participants were enrolled in the same villages.

Many thanks for this suggestion. We agree that adequate control of relatedness in this analysis is very important. To address this, in order to confirm that the observed association between rs13390936 and NTS bacteraemia is not confounded by sample relatedness, we have reanalysed that association using a linear mixed model, using all Kenyan discovery samples (regardless of pairwise IBD), to account for population structure and relatedness. In that analysis, we confirm that the association between rs13390936 genotype and NTS bacteraemia is independent of relatedness in the Kenyan discovery sample collection. This analysis is presented in the main text as follows:

“Re-analysis of the STAT4 locus in the Kenyan discovery samples (181 cases, 2,759 controls; this includes samples excluded from the main analysis for relatedness) using a linear mixed model, to account for population structure and relatedness, confirms the association of rs13390936 genotype with NTS bacteraemia (recessive model; $P = 7.14 \times 10^{-7}$).”

We detail the associated methods as follows:

“Re-imputed genotypes were further used to test for association at STAT4, under a recessive model of association, using a linear mixed model to account for population structure and sample relatedness in GEMMA³⁷. In this analysis, samples were not excluded according to pairwise relatedness, but QC thresholds were otherwise as described for the Kenyan discovery samples.”

Reviewers' comments:

Reviewer #1 (Remarks to the Author):

My comments have been addressed; I have no further issue.

Reviewer #2 (Remarks to the Author):

The authors clarified all the requested points and their revision made the association of an intronic variant of STAT4 gene with NTS bacteraemia more reliable. Now I agree to accept their manuscript.

Minor points;

1, Page7, Line117: Add 95% Confidence Interval for OR.

2, Page12, Line220: Refer to Supplementary Fig. 9 (not 8)

Reviewer #3 (Remarks to the Author):

Malaria co-infection is known to be a significant risk factor for NTS, and the presented fitted regression model including malaria co-infection shows a trend of association ($p = 0.07$) in the Malawian group. As there is overwhelming data documenting the influence of malaria on NTS infection all association models should correct for the presence of malaria in all analysed study groups. In addition, no analyses for the influence of risk factors were presented for the Kenyan discovery group. This must also be made up to exclude a possible bias in this group.

Reviewer Responses

Risk of nontyphoidal *Salmonella* bacteraemia in African children is modified by *STAT4*

Reviewer #1 (Remarks to the Author):

My comments have been addressed; I have no further issue.

Many thanks for taking the time to review our work.

Reviewer #2 (Remarks to the Author):

The authors clarified all the requested points and their revision made the association of an intronic variant of *STAT4* gene with NTS bacteraemia more reliable. Now I agree to accept their manuscript.

Many thanks for taking the time to review our revised manuscript.

Minor points;

1, Page7, Line117: Add 95% Confidence Interval for OR.

The 95% confidence interval has been added.

2, Page12, Line220: Refer to Supplementary Fig. 9 (not 8)

Many thanks for this, this has been amended.

Reviewer #3 (Remarks to the Author):

Malaria co-infection is known to be a significant risk factor for NTS, and the presented fitted regression model including malaria co-infection shows a trend of association ($p = 0.07$) in the Malawian group.

Many thanks for highlighting that this was not adequately clear. The p-value of 0.07 is derived from a likelihood ratio test of models of NTS bacteraemia with and without interaction terms between rs13390936 genotype and malaria. A significant result would suggest interaction/effect modification, rather than confounding or bias (which is addressed in the preceding model adjusted for acquired risk factors including malaria). To make this clearer we have amended the relevant results section to now read (changes highlighted):

“While there is no evidence that the observed association between rs13390936 genotype and NTS bacteraemia is the result of confounding secondary to acquired risk factors for NTS, we further investigated whether there is evidence that the effect of rs13390936 genotype on NTS disease risk is modified by HIV, malaria or malnutrition. We fitted regression models of NTS bacteraemia and rs13390936 association, with and without interaction terms between rs13390936 and each acquired risk factor, in the same set of Malawian children. In that analysis, there is no evidence for interaction between carriage of the rs13390936:TT genotype and HIV status ($P=0.81$), malaria co-infection ($P=0.07$) or malnutrition ($P=0.82$) in NTS bacteraemia risk.”

We have also amended the discussion as follows:

“It is important to recognise, that while our analysis provides no evidence for acquired NTS risk factors modifying the effect of rs13390936 genotype on risk of NTS disease, this study has limited power to detect such an interaction. It will therefore be important to establish in future studies, whether the association between rs13390936 and risk of NTS bacteraemia is observed consistently across populations of African children with diverse rates of HIV, malaria and malnutrition.”

As there is overwhelming data documenting the influence of malaria on NTS infection all association models should correct for the presence of malaria in all analysed study groups. In addition, no analyses for the influence of risk factors were presented for the Kenyan discovery group. This must also be made up to exclude a possible bias in this group.

Many thanks for raising this. As we have highlighted in the manuscript, the fitting of regression models adjusting for acquired risk factors is not possible in Kenyan discovery cohort as the control samples were recruited as a birth cohort. We now emphasise this point in the manuscript as follows:

“Control samples used in the Kenyan discovery analysis are taken from a birth cohort, and acquired risk factor data are not available to perform a regression analysis of NTS bacteraemia risk adjusted for malaria, malnutrition and HIV.”

To address this limitation, we now provide additional analysis in the Kenyan discovery cohort, performing a Bayesian comparison of models of association at rs13390936 with NTS bacteraemia with and without acquired risk factors for NTS. This analysis is presented in a new supplementary figure 4, and demonstrates that there is again no evidence that the observed association of rs13390936 genotype with NTS bacteraemia is secondary to confounding or bias due to malaria co-infection or malnutrition. We describe this analysis in the manuscript as follows (changes highlighted):

“Malaria, HIV infection and malnutrition are important, acquired risk factors for NTS bacteraemia in African children^{6,7}. To address whether the observed association of rs13390936 with NTS bacteraemia is independent of these acquired risk factors, we conducted a Bayesian analysis comparing models of association at rs13390936 with NTS bacteraemia with and without acquired risk factors for NTS in the Kenyan discovery samples (Supplementary Fig. 4). In that analysis, the most probable models are those in which rs13390936 is associated with susceptibility to NTS bacteraemia with the same effect size in children with and without each of malaria and malnutrition.”

The small numbers of HIV-infected children (n=24) with NTS bacteraemia in the Kenyan discovery cohort do not permit a stratified analysis. To exclude confounding by HIV status among the Kenyan discovery cases, we now repeat the association analysis at rs13390936 including only HIV-uninfected cases. This analysis demonstrates an unchanged effect size estimate (as compared to the overall analysis). These changes are detailed in the manuscript as follows:

“The numbers of HIV-infected children in the Kenyan discovery samples (n=24, none of which carry the rs13390936:TT genotype) are too small to permit a stratified analysis. To ensure HIV infection did not confound the observed association between rs13390936 genotype and NTS bacteraemia, we repeated the association analysis at rs13390936 including only HIV-uninfected NTS cases (n = 97). In that analysis, the observed effect size (recessive model; OR 8.49, 95% CI 2.37-24.28) is in keeping with that observed in the association analysis including all samples (Table 1), further supporting the association between rs13390936 genotype and NTS bacteraemia.”

Importantly, previously published, well-powered GWAS of multiple HIV and malaria-related traits have demonstrated no evidence for association between these phenotypes and genetic variation at *STAT4*. This, in addition to the lack of evidence for confounding in the data presented in this study, strongly argues against the observed association at rs13390936 with NTS bacteraemia being secondary to confounding. We now highlight this lack of evidence in the manuscript, as follows:

*“This lack of confounding by acquired risk factors for NTS is in keeping with the absence of observed association between genetic variation at the *STAT4* locus and HIV or malaria-related phenotypes in GWAS published to date^{32,33}.”*

We further specifically explore whether genotype at rs13390936 is associated with malaria in Kenyan and Malawian children, using publically available data generated by the MalariaGEN consortium. In that dataset, there is no evidence for association between rs13390936 genotype and severe malaria under any genetic model. This analysis is detailed in the manuscript as follows:

“To further address the possibility of confounding secondary to malaria in Kenyan and Malawian children, we used publically available GWAS summary statistics from the MalariaGEN consortium GWAS of severe malaria in Gambian, Kenyan and Malawian children¹² (total 5,126 cases, 5,287 controls; Gambia 2,428 cases, 2,489 controls; Malawi 1,193 cases, 1,321 controls; Kenya 1,505 cases, 1,477 controls) to assess for any evidence of association between rs13390936 and severe malaria. In that dataset, rs13390936 is not associated with severe malaria in African children (additive model, $P=0.31$, OR 0.96, 95% CI, 0.88-1.04; dominant model $P=0.47$, OR 0.97, 95% CI, 0.88-1.06; recessive model $P=0.15$, OR 0.76, 95% CI, 0.52-1.11; heterozygous advantage model $P=0.98$, OR 0.98, 95% CI, 0.89-1.08).”

REVIEWERS' COMMENTS:

Please note that while Reviewer 3 doesn't have Remarks to the Author, in his/her Remarks to the Editor, he/she said the authors clarified the unclear points, he/she agrees to accept the manuscript.